METHODS AND RESOURCES

# Nondisruptive inducible labeling of ER-membrane contact sites using the Lamin B receptor

**Laura Downie, Nuria Ferrandiz¤, Elizabeth Courthold, Megan Jones, Stephen J. Royle** *

Centre for Mechanochemical Cell Biology, Warwick Medical School, University of Warwick, Coventry, United Kingdom

¤ Current address: Centro de Investigación del Cáncer, Salamanca, Spain
* s.j.royle@warwick.ac.uk

## Abstract

Membrane contact sites (MCSs) are areas of close proximity between organelles that allow the exchange of material, among other roles. The endoplasmic reticulum (ER) has MCSs with a variety of organelles in the cell. MCSs are dynamic, responding to changes in cell state, and are, therefore, best visualized through inducible labeling methods. However, existing methods typically distort ER-MCSs, by expanding contacts or creating artificial ones. Here, we describe a new method for inducible labeling of ER-MCSs using the Lamin B receptor (LBR) and a generic anchor protein on the partner organelle. Termed *LaBeRling*, this versatile, one-to-many approach allows labeling of different types of ER-MCSs (mitochondria, plasma membrane, lysosomes, early endosomes, lipid droplets, and Golgi), on-demand, in interphase or mitotic human cells. LaBeRling is nondisruptive and does not change ER-MCSs in terms of the contact number, extent or distance measured; as determined by light microscopy or a deep-learning volume electron microscopy approach. We applied this method to study the changes in ER-MCSs during mitosis and to label novel ER-Golgi contact sites at different mitotic stages in live cells.

## Introduction

In eukaryotic cells, membrane contact sites (MCSs) are areas of close proximity between two membranes of different identity. MCSs allow the exchange of material between the two membranes without fusion, and they also regulate organelle positioning, dynamics, and number [1]. The ER occupies a large volume of the cell and forms contacts with multiple membranes of different identity [2]. ER-MCSs, therefore, play a central role in cellular communication across organelles; they are also dynamic and must adapt in response to changes in cell state. The ER and other membrane compartments remodel upon entry into mitosis [3]. For example, the nuclear envelope breaks down and the Golgi fragments [4,5]. How ER-MCSs change in response

**Data availability statement:** All code and data used to generate the plots in the manuscript are available at https://github.com/quantixed/p065p038 and archived at Zenodo (https://doi.org/10.5281/zenodo.15582238) with additional data available at Zenodo (https://doi.org/10.5281/zenodo.11396014).

**Funding:** This work was supported by a Programme Award from Cancer Research UK (C25425/A27718 to SJR), studentships from BBSRC Midlands Integrative Biosciences Training Partnership (MIBTP2, BB/M01116X/1 to LD and MIBTP2020, BB/T00746X/1 to MJ), and a grant from the Human Frontier Science Program (HFSP; RGP25/2022 to SJR). The funders had no role in study design, data collection and analysis, decision to publish, or preparation of the manuscript.

**Competing interests:** The authors have declared that no competing interests exist.

**Abbreviations:** ddFP, dimerization-dependent FP; EDT, Euclidean distance transform; EM, electron microscopy; ER, endoplasmic reticulum; FKBP, FK506 binding protein; FP, fluorescent protein; FRET, fluorescence resonance energy transfer; GFP, green fluorescent protein; LAMP1, lysosome-associated membrane glycoprotein 1; LBR, Lamin B receptor; MCS, membrane contact site; PB, phosphate buffer; PLIN3, perilipin-3; PM, plasma membrane; SBF-SEM, serial block face-scanning electron microscopy; TGN, trans-Golgi network; TM, transmembrane.

to this remodeling is not well understood. Recent work indicates that changes in ER-MCSs may be coordinated with other mitotic processes through regulation of tethering proteins in mitosis [6]. The study of MCSs is hampered by the lack of live cell labeling methods that (i) specifically label MCSs, (ii) do not interfere with contacts, and (iii) allow visualization of contacts during specific cell cycle stages.

ER-MCSs were first observed in electron microscopy (EM) studies in fixed cells [1,7–10]. Other visualization methods include those based on measures of organelle membrane proximity from live cell multispectral imaging [11] or fixed super-resolution 3D images [12]. Ideally, MCS labeling methods in live cells must distinguish MCS from regions where membranes are in close proximity by chance, as recently reviewed by Nakatsu and Tsukiji [13]. Examples for visualizing ER-MCSs include proximity-dependent methods, where fluorescence is produced when protein tags on each membrane are in close proximity: fluorescence resonance energy transfer (FRET) [14], split fluorescent proteins (FPs) [15,16], and dimerization-dependent FPs (ddFPs) [17]. Synthetic constructs based on tethering proteins have also been used to visualize ER-MCSs [18]. A major limitation of these methods is that labeling is not inducible. To study MCSs during mitosis, labeling that persists and potentially stabilizes interphase contacts is undesirable.

Inducible methods allow temporal control of MCS labeling and typically use tagged proteins on each apposing membrane that heterodimerize in response to light or a chemical [13,19–21]. However, the initial expression and/or the induced heterodimerization of these proteins usually increase the contact between membranes, through expanding the area of pre-existing MCSs or by creating artificial tethers between the ER and the apposing membrane. Expansion of ER-mitochondria MCSs over time after inducing the labeling has been observed in live cells and by EM [22,23]. Here, the ER could be seen wrapping around the mitochondria surface; increasing ER-mitochondria coverage from ~10% to ~30%–90% after induction [22]. Moreover, an optogenetic approach also showed an increase of ~20% in ER/mitochondria signal overlap in live HeLa cells; and a larger increase in ER-lysosome contacts using this approach in live Cos-7 cells [24]. ER-plasma membrane (PM) contacts were also increased after labeling using rapamycin-induced heterodimerization [25]. In fact, we previously exploited this property to artificially "glue" the ER to the PM during mitosis to free chromosomes trapped by the ER [26]. This limits the use of inducible methods for the study of MCSs, as labeling manipulates the ER-MCSs. Furthermore, inducible methods that are less disruptive rely on expensive proprietary reagents prompting the need for the development of new ER-MCS labeling methods [20,27].

Our aim was to develop an inducible labeling system which allows fast and specific labeling of ER-membrane contacts, without disrupting the contacts themselves. Using chemical-induced heterodimerization with a tagged anchor protein at the target membrane, we serendipitously found that Lamin B receptor (LBR), which localizes to the ER and inner nuclear envelope, specifically labeled ER-MCSs upon relocalization. We called this method *LaBeRling*. We found that LaBeRling caused no detectable change in the contact number, extent or distance measured from high-resolution

3D EM datasets. It can be used to label multiple different ER-MCS types, and we applied this method to study MCSs in mitosis and use it to reveal novel ER-Golgi MCSs in live mitotic cells.

## Results

### LBR forms clusters after relocalization to the PM

In principle, induced heterodimerization of proteins tagged with FK506 binding protein (FKBP) and FRB domains can be used to label MCSs on-demand. The advantage to this method is that the moment of labeling is controlled by the investigator, a major disadvantage is that the heterodimerization may distort existing contact sites or even induce new, or artificial contacts (Fig 1A). We began by comparing the relocalization of two different proteins: Sec61β and LBR; to the PM using induced heterodimerization. Both Sec61β and LBR are found in the ER, with LBR being additionally present on the inner nuclear envelope in interphase. Using Stargazin-dCherry-FRB as a PM anchor, we found that FKBP-GFP (green fluorescent protein)-Sec61β relocalization causes the ER to become glued to the PM in interphase or during mitosis, as reported previously [26]. By contrast, LBR-FKBP-GFP was found in discrete clusters at the PM following rapamycin addition (Fig 1B). The formation of LBR-FKBP-GFP clusters was dependent on both the expression of the PM anchor and the addition of rapamycin (S1 Fig).

To avoid the possibility that overexpression of LBR-FKBP-GFP contributed to cluster formation, we generated a knock-in cell line where LBR was tagged with FKBP-GFP at its endogenous locus (S2 Fig). Using these cells, we confirmed that the formation of LBR-FKBP-GFP clusters occurred in a similar way (S1 Fig and S1 Video). Furthermore, similar LBR-FKBP-GFP clusters were seen when using SH4-FRB-EBFP2, a peripheral membrane protein, in place of the multipass Stargazin PM anchor (S3 Fig), which suggested that the identity of the anchor was unimportant and that the difference in behavior could be attributed solely to the ER-resident protein. Interestingly, there was minimal clustering of the PM anchor upon relocalization of LBR-FKBP-GFP (Figs 1C and S3A). We also found no evidence for co-clustering of mCherry-Sec61β or of LBR-mCherry when LBR-FKBP-GFP was relocalized to the PM, which suggested that the clusters did not represent nonspecific aggregates of ER or LBR protein itself (S3 Fig). Visualization of the ER using PhenoVue Fluor568-ConA or mCherry-Sec61β revealed no gross changes in ER morphology when clustering of LBR-FKBP-GFP was induced (S4 Fig and S2 and S3 Videos). Indeed, because the ER stayed intact while LBR-FKBP-GFP clustered in the ER at discrete sites on the PM, it suggested that this manipulation may be labeling ER-PM contact sites.

### Properties of LBR-FKBP-GFP clusters at the PM following relocalization

We next investigated the properties of the clusters that form after inducing the relocalization of LBR-FKBP-GFP to the PM. We characterized the properties of the LBR-FKBP-GFP clusters in cells in interphase or at metaphase using 3D segmentation of confocal z-stacks of HCT116 LBR-FKBP-GFP knock-in cells expressing Stargazin-mCherry-FRB treated with rapamycin (200 nM, 20 min) (Fig 1C). The surface area of the clusters (see Methods for definition) was variable, but the median cluster area per cell was (0.50 ± 0.07, interphase; 0.55 ± 0.06 μm$^2$, metaphase), with no significant difference between interphase and metaphase cells (Fig 1D and 1E). Hundreds of clusters were detected in each cell, but to normalize for differences in cell size and understand the density of LBR-FKBP-GFP clusters at the PM a cell surface approximation was generated and the density of clusters per unit area was determined (Fig 1F and 1G). Again, the density of clusters was similar in interphase and metaphase cells (0.26 ± 0.12, interphase; 0.21 ± 0.08 μm$^{-2}$, metaphase). Given the average size of the clusters and their density, the coverage of the PM was ~10%, which is similar to published estimates of PM coverage with ER-PM contact sites [28].

To investigate the formation and dynamics of LBR-FKBP-GFP clusters, we imaged mitotic cells during the induced relocalization of LBR-FKBP-GFP to the PM (Fig 1H). These movies revealed that multiple clusters that were distributed around the cell, formed simultaneously and with similar kinetics (S1 Video). The clusters began to form ~90 s after rapamycin addition, with typically less than 20 clusters in a single confocal slice, and persisted throughout the duration of the

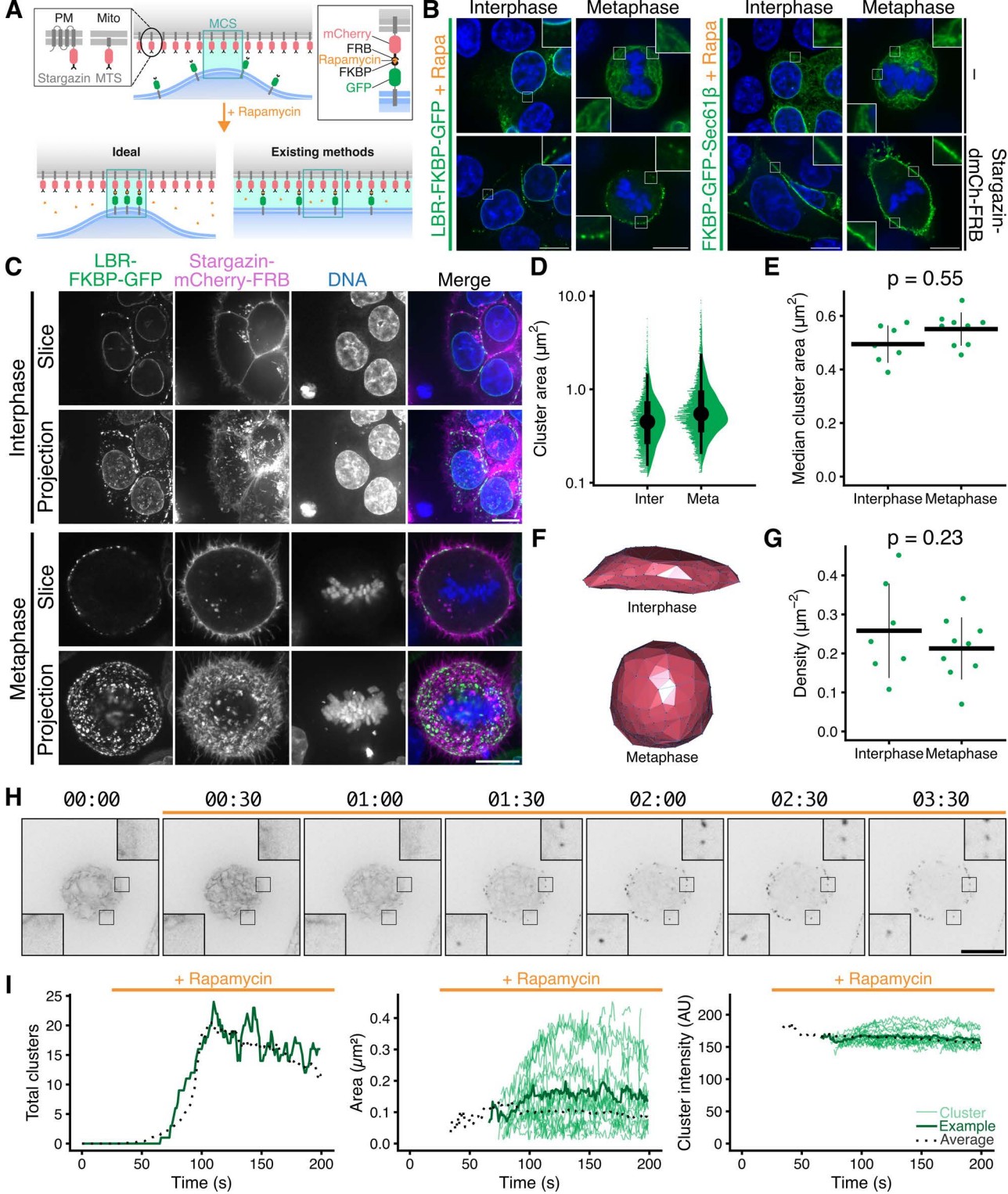

**Fig 1. Properties of LBR-FKBP-GFP clusters at the plasma membrane following induced relocalization. (A)** Schematic diagram to show heterod-imerization of FKBP and FRB domains with rapamycin and how this may be used to label ER-membrane contact sites but also how this method may distort contact sites. Membrane contact site (MCS) is labeled by a blue box and a green shading is used indicate membranes within close distance of

each other. **(B)** Example micrographs of HCT116 cells co-expressing LBR-FKBP-GFP or FKBP-GFP-Sec61β (green) and optionally Stargazin-dCherry-FRB, each treated with rapamycin (200 nM) for 30 min before fixation and DNA staining (blue). Scale bars, 10 μm; Insets, 3× expansion of ROI. Note: Additional controls are shown in full in S1A Fig. **(C)** Micrographs of typical HCT116 LBR-FKBP-GFP (green) knock-in cells expressing Stargazin-mCherry-FRB (magenta), stained with SiR-DNA (blue), treated with rapamycin (200 nM). Single confocal slices or z-projections for cells in interphase or mitosis are shown. **(D)** Raincloud plot to show size distribution of LBR-FKBP-GFP clusters analyzed in 3D. **(E)** Plot of the median contact area for each cell (dots). Mean and ± sd are indicated by crossbar. **(F)** 3D cell surface approximation generated using the location of all segmented LBR-FKBP-GFP clusters (see Methods). (G) Plot of the density of clusters (total clusters divided by cell surface). Dots show cells, mean and ± sd are indicated by cross-bar. **(H)** Stills from a movie of LBR-FKBP-GFP cluster formation upon rapamycin addition (200 nM, orange bar). Cells were as described in C. Timescale, mm:ss. Scale bars, 10 μm; Insets, 3× expansion of ROI. **(I)** Plots to show the number, size, and intensity of LBR-FKBP-GFP clusters in a single slice over time. Thin green lines and dark green line, clusters from and average for the cell shown in F; black dotted line, average of 14 different cells. The individual values for panels D, E, G, and I are available at https://doi.org/10.5281/zenodo.15582238.

movie (up to 5 min). The clusters that formed did so at the expense of fluorescence in the ER which "drained away" with similar kinetics.

Using a spot-tracking procedure, we could monitor the behavior of each cluster. Analysis revealed that once formed, the total number, size, and brightness of the clusters were essentially constant on this timescale (Fig 1I). There were very few splitting or merging of clusters, and any appearance or disappearance of clusters could be attributed to movement into or out of the imaging plane. Overall, the mobility of the clusters was very low (0.016 μm s$^{-1}$, median, $n = 16$ cells).

The coordinated appearance and stability of fluorescence suggests that relocalized LBR-FKBP-GFP labels pre-existing ER-PM contact sites.

### Relocalized LBR-FKBP-GFP labels pre-existing ER-PM contact sites

Are the LBR-FKBP-GFP clusters that form after relocalization, coincident with pre-existing ER-PM contact sites? To answer this question, we took confocal z-stacks of interphase and mitotic cells before and after LBR-FKBP-GFP relocalization to Stargazin-EBFP2-FRB using rapamycin (200 nM). To identify the ER-PM contact sites, we used a modified MAPPER construct (mScarlet-I3-6DG5-MAPPER), which is an established marker of ER-PM contact sites [18]. From these images, we could clearly see colocalization between the contact sites marked by MAPPER and the clusters of relocalized LBR-FKBP-GFP in interphase and mitotic cells (Fig 2A). This was also true of the relocalization captured in live HCT116 or HeLa cells (S4 and S5 Videos). The contact sites where colocalization occurred were present before LBR-FKBP-GFP was relocalized, indicating that labeling is of pre-existing contact sites.

3D segmentation and quantification of the contact sites marked by MAPPER and the clusters of LBR-FKBP-GFP confirmed that the formation of LBR-FKBP-GFP clusters by inducing relocalization was significant. There was no change in the number of MAPPER clusters per cell in mitotic cells, and a small but significant decrease in interphase which was probably attributable to photobleaching (Fig 2B). Importantly, we found no evidence for increases in MAPPER clusters after LBR-FKBP-GFP relocalization, which would have indicated the artificial formation of new contact sites. When the colocalization of the two fluorescence channels was measured we found in most cells LBR-FKBP-GFP clusters were contact sites (Fig 2C). The fraction of contact sites that were labeled by LBR-FKBP-GFP was lower, a result which may have been influenced by photobleaching of GFP versus mScarlet-I3.

We found no influence of MAPPER expression on the number of LBR-FKBP-GFP clusters that form upon relocalization to the PM. For example, there were $213.4 \pm 89.4$ LBR-FKBP-GFP clusters in MAPPER expressing interphase cells compared with $277.2 \pm 112.6$ in those not co-expressing MAPPER ($p = 0.7$, Tukey's post-hoc test). This indicates that there is no interference between the two labeling types.

Finally, this dataset also revealed that there are fewer ER-PM contact sites at metaphase than there are in interphase (Fig 2B). For example, the number of MAPPER clusters in interphase and metaphase cells, before rapamycin treatment was significantly lower ($429.3 \pm 74.5$, interphase; $258.5 \pm 45.3$, metaphase; $p = 0.002$). A pattern repeated for post-rapamycin treatment ($p = 0.002$) and for LBR-FKBP-GFP clusters in MAPPER expressing cells post-rapamycin ($p = 0.04$).

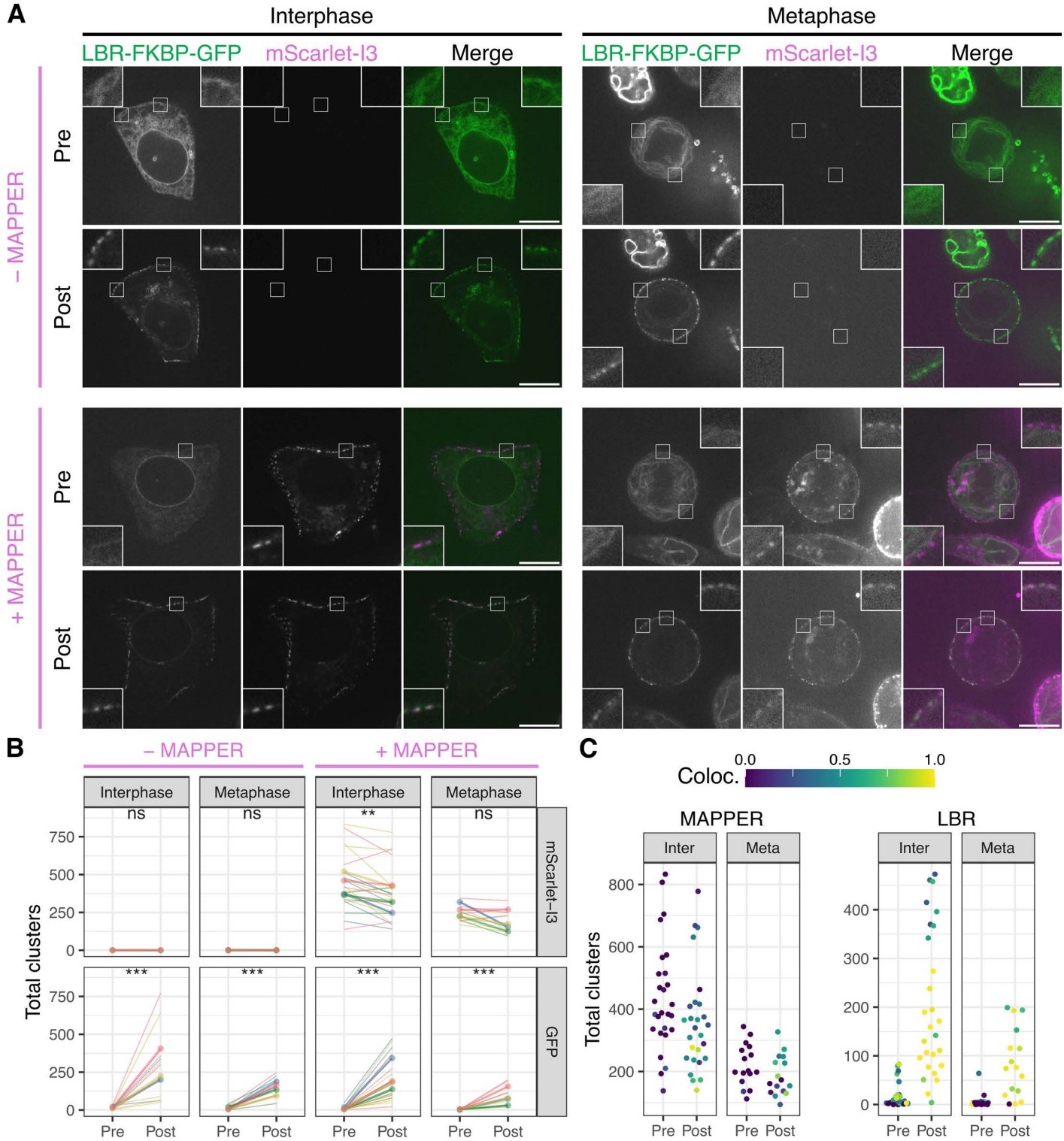

**Fig 2. Relocalized LBR-FKBP-GFP labels pre-existing ER-PM contact sites. (A)** Example micrographs of HCT116 cells co-expressing LBR-FKBP-GFP (green) and Stargazin-EBFP2-FRB and optionally MAPPER (mScarlet-I3-6DG5-MAPPER, magenta) as indicated. A single slice of a z-stack of the same cell is shown before (pre) or after (post) rapamycin (200 nM, 20 min) addition. Scale bars, 10 µm; Insets, 3× expansion of ROI. **(B)** Comparison of total 3D clusters per cell detected in mScarlet-I3 (MAPPER) or GFP (LBR-FKBP-GFP) channels. Thin lines indicate the pre and post values for each cell.

Thick lines and dots indicate the average per experimental repeat. Color indicates experimental repeat. Paired *t* tests with Holm–Bonferroni correction for multiple testing: ns, not significant; *** $p < 0.001$; ** $p < 0.01$. **(C)** Colocalization analysis. For cells co-expressing MAPPER, the number of MAPPER or LBR clusters is shown and the extent that these clusters colocalize with LBR or MAPPER clusters is indicated by the colorscale. The individual values for panels B and C are available at https://doi.org/10.5281/zenodo.15582238.

In summary, we could confirm that LBR-FKBP-GFP relocalization to the PM labels pre-existing ER-PM contact sites and that there were no additional sites created by this relocalization. We also documented a decrease in the number of contacts in mitotic cells compared with nondividing cells.

### Using relocalization of LBR-FKBP-GFP to detect functional changes in ER-PM membrane contact sites

Application of the ER calcium pump inhibitor, thapsigargin, was reported to increase the density of ER-PM MCSs [18]. To test if the relocalization of LBR-FKBP-GFP to the PM was sensitive to functional changes in MCSs, we applied thapsigargin (1 µM) for 20 min prior to application of rapamycin. Measurement of the resulting LBR-FKBP-GFP clusters showed a significant increase in coverage at the PM (S5 Fig). These experiments suggest that the relocalization of LBR-FKBP-GFP to the PM can be used to report changes in MCS size when ER calcium signaling is manipulated.

### Relocalization of LBR-FKBP-GFP to mitochondria highlights ER-mitochondria contact sites

Having established that LBR can be used to inducibly label ER-PM contact sites, we next tested if it could be used to label ER-mitochondria contact sites. HCT116 cells transiently co-expressing MitoTrap (Mito-mCherry-FRB) and either LBR-FKBP-GFP or FKBP-GFP-Sec61β were imaged in interphase or mitosis (Fig 3A). Following relocalization with rapamycin (200 nM), LBR-FKBP-GFP was clustered at discrete sites on each mitochondrion with typically one contact per mitochondrion visible by light microscopy. By contrast, FKBP-GFP-Sec61β fluorescence completely surrounded each mitochondrion suggesting that the relocalization of this protein caused the ER to wrap around the mitochondria (Fig 3A). This pattern was similar in interphase or mitotic cells, and mirrored the observations previously using a PM anchor.

The clusters of LBR-FKBP-GFP on mitochondria following relocalization were also observed with the endogenously tagged protein in live cells, again suggesting that the cluster formation was not an artifact of overexpression or fixation (Fig 3B). Live-cell imaging of the relocalization revealed that cluster formation was rapid (~20 s) and came at the expense of fluorescence in the ER (S6 Video). Wrapping of mitochondria by FKBP-GFP-Sec61β occurred on longer timescale (S7 Video). We used fluorescence profiles to determine the fraction of the organelle that was occupied by GFP signal (Fig 3C), this analysis revealed significantly higher coverage by FKBP-GFP-Sec61β compared to LBR-FKBP-GFP (Fig 3D).

The appearance of discrete clusters of LBR-FKBP-GFP upon relocalization to mitochondria suggested that these clusters most likely represent ER-mitochondria contact sites.

### Inducible labeling of ER-mitochondria contacts using LBR does not affect ER-mitochondria contacts

Are ER-mitochondria contacts altered by the relocalization of LBR-FKBP-GFP to MitoTrap? As there was no obvious choice of marker to assess this, we used a 3D-EM approach to examine all ER-mitochondria contacts (Fig 4). We first imaged live HCT116 LBR-FKBP-GFP knock-in cells co-expressing MitoTrap and confirmed the relocalization of LBR-FKBP-GFP to mitochondria or not in the case of the control (no rapamycin, Fig 4A). The same cell was then processed for serial block face-scanning electron microscopy (SBF-SEM) and relocated for imaging by 3D-EM. We used a machine learning approach to infer in 3D the mitochondria and ER in all of the resulting datasets (control, 3; rapamycin, 4) using a manually segmented subvolume from one dataset (Fig 4B). Next, using the inferred maps of mitochondria and ER, we used an automated procedure to detect the regions of ER that were within a defined distance of the mitochondria and then segment those to measure the size of the ER region that is in contact with the mitochondrion (see Methods).

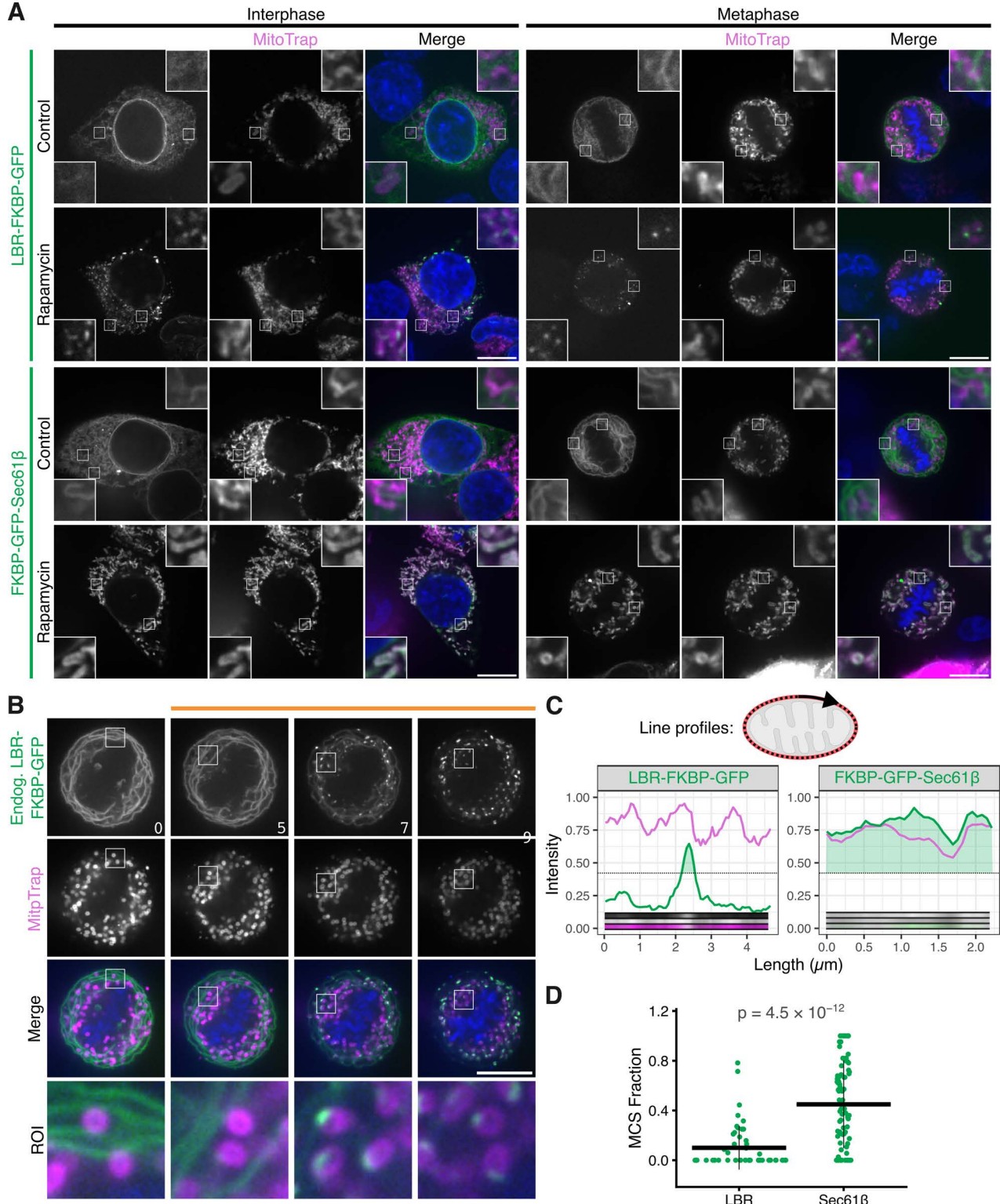

**Fig 3. Relocalization of LBR-FKBP-GFP to mitochondria highlights ER-mitochondria contact sites. (A)** Example micrographs of HCT116 cells expressing either LBR-FKBP-GFP or FKBP-GFP-Sec61β (green) and MitoTrap (Mito-mCherry-FRB, magenta). Relocalized samples were treated

with rapamycin (200 nM) for 30 min before fixation. Control samples were not treated with rapamycin. A single slice from a z-stack of an interphase or metaphase cell are shown. Contrast of the MitoTrap channel was adjusted for clarity. Scale bars, 10 μm; Insets, 4× expansion of ROI. **(B)** Stills from a live cell imaging experiment with HCT116 LBR-FKBP-GFP (green) knock-in cells co-expressing MitoTrap (magenta), treated with rapamycin (200 nM) as indicated. Time, min; Scale bars, 10 μm; Zooms, 6.5× expansion of ROI. **(C)** Line profiles measured around the mitochondrial perimeter from mitotic cells in A, as represented in the schematic. Plots show the intensity of LBR-FKBP-GFP or FKBP-GFP-Sec61bβ (green) and MitoTrap (magenta) signal measured around an individual mitochondria. Dotted line indicates the threshold for segmentation, and light green area indicates sections of the line above the threshold. Insets show the line profile images. **(D)** Plot of the fraction of the mitochondrial perimeter that is above the threshold. Quantification is of line profiles like those shown in C. *P*-value, Tukey's HSD *post-hoc* test; $n_{cell}$ = 3 (LBR), 8 (Sec61β). The individual values for panels C and D are available at https://doi.org/10.5281/zenodo.15582238.

We verified that the procedure identified genuine ER-mitochondrion contacts by sampling a number of contacts and by visualizing them in 3D (Fig 4B). Using the resulting data, we could measure the total ER contacts per mitochondrion and plot the total contact area as a function of mitochondrion surface area (Fig 4C). At all defined distances assessed, from 10 to 50 nm, we found that the contacts were similar between the two experimental groups (Fig 4D). These data indicate that relocalization of LBR-FKBP-GFP can be used to discretely label ER-mitochondria contact sites, without affecting the morphology of those contacts. Since LBR labeling of ER-PM contacts was shown to be similarly noninvasive, we propose that our method is an innocuous way to label contact sites inducibly. We call this method, LaBeRling.

### General application of the LaBeRling method

Having demonstrated LaBeRling in HCT116 cells, we wanted to test if the method worked in other cell types. We tested LaBeRling of ER-PM and ER-mitochondria MCSs in three cell lines: RPE-1, Cos-7, and HeLa. In each case, relocalization of LBR-FKBP-GFP to its respective anchor using rapamycin (200 nM) induced discrete puncta formation in each cell type (S6 Fig). This suggests that the method may be broadly applicable in a variety of cell lines.

So far, we have used up to 30 min rapamycin addition to induced LaBeRling. For most experiments, this will be sufficient; however, for long-term highlighting of MCSs, the use of rapamycin as an induction agent may be problematic due to its bioactivity on the timescale of hours. Accordingly, we tested whether rapalog AP20187 could similarly be used to induce LaBeRling. Using coexpression of either Stargazin-mCherry-FRB(T2098L) or Mito-mCherry-FRB(T2098L) variants with LBR-FKBP-GFP, we saw similar highlighting of ER-PM or ER-mitochondria MCSs, respectively (S7 Fig).

We next attempted long-term LaBeRling of ER-PM MCSs in HeLa cells. Following induction, we could see that contacts remained brightly fluorescent up to 4 h (S8 Fig). Therefore, we tested whether such long-term LaBeRling was detrimental to cell health. Using this method, we could follow cells with LaBeRled ER-PM MCSs for up to ~9 h. Cells migrated similarly to their respective controls and the MCSs remained fluorescent throughout (S8 and S9 Videos). These observations suggest that the method can be used in other contexts beyond short-term highlighting of contact sites, and that LaBeRling is effective in other cell lines.

### Relocalized LBR-FKBP-GFP can mark several different ER-membrane contact types

As LaBeRling can be used to highlight ER-PM and ER-mitochondria contact sites, we next asked if it could be applied to similarly label other ER-MCSs. To do this, we tested three additional anchor proteins: perilipin-3 (PLIN3), early endosome antigen 1 (EEA1), and lysosome-associated membrane glycoprotein 1 (LAMP1); that mark lipid droplets, early endosomes, and lysosomes, respectively. We compared the relocalization of LBR-FKBP-GFP with that of FKBP-GFP-Sec61β to differentiate genuine contact site labeling from nonspecific recruitment of ER to the target membrane.

At early endosomes and lysosomes, LBR-FKBP-GFP was relocalized to clusters, whereas the relocalization of FKBP-GFP-Sec61β matched the fluorescence of the anchor (Fig 5A). At lipid droplets, however, relocalization of either LBR-FKBP-GFP or FKBP-GFP-Sec61β caused a coincidence of the GFP fluorescence with FRB-mCherry-PLIN3 that was not distinguishable between the two ER proteins (Fig 5A). Analysis of the MCS fraction from organelle fluorescence profiles

PLOS Biology

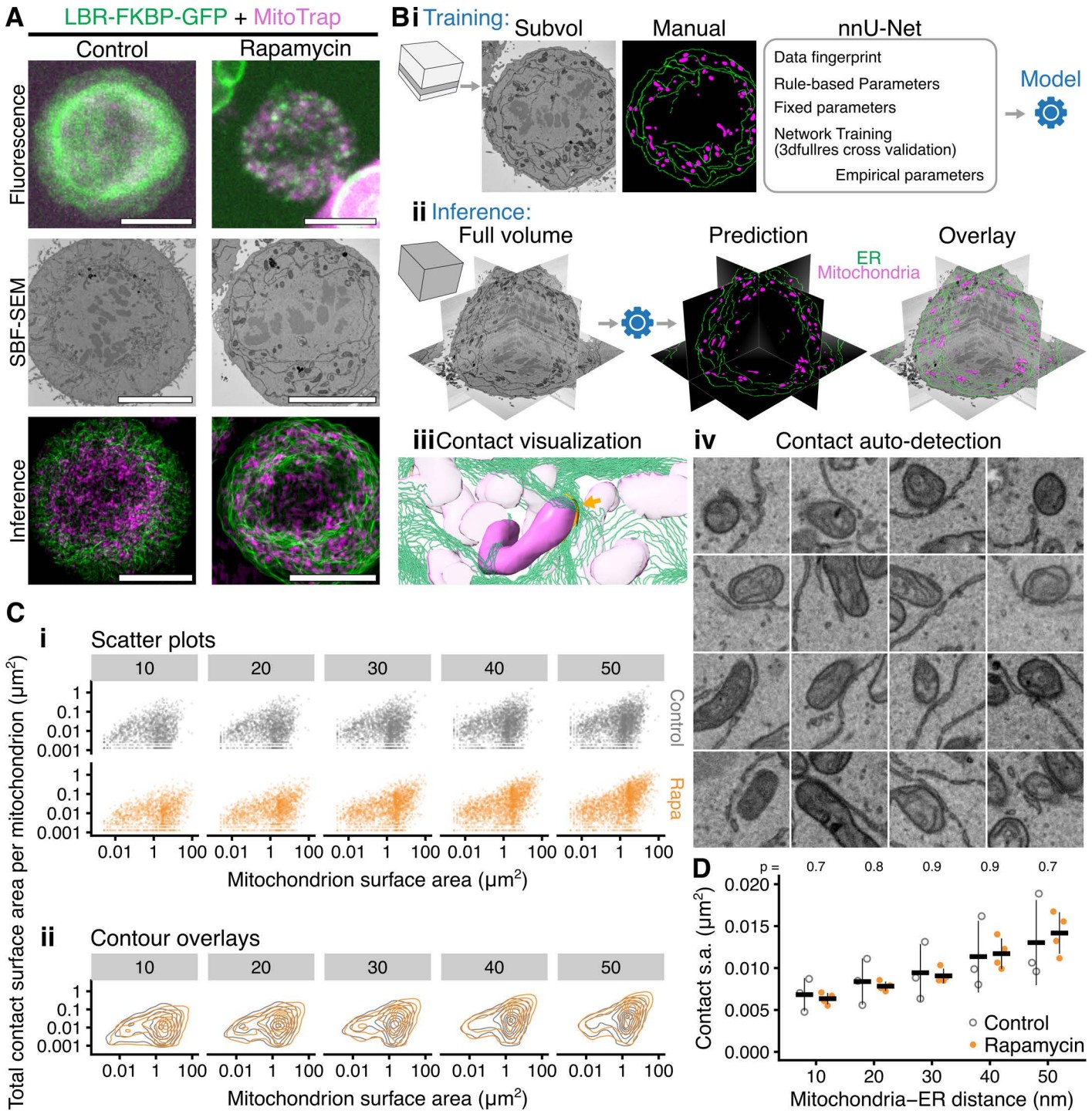

**Fig 4. Inducible labeling of ER-mitochondria contacts using LBR does not affect ER-mitochondria contacts. (A)** Example micrographs of HCT116 LBR-FKBP-GFP (green) knock-in cells expressing MitoTrap (magenta) treated with Rapamycin (200 nM, 30 min) or not (Control). Mitotic stage was confirmed by imaging SiR-DNA. Following fixation and processing, the same cell was imaged again by serial block face-scanning electron micros-copy (SBF-SEM). Finally, SBF-SEM datasets were used to infer the location of ER (green) and mitochondria (magenta), a Z-projection is shown. Scale bar, 10 μm. **(B)** Large-scale machine learning segmentation of ER and mitochondria from SBF-SEM data. (i) Supervised training of nnU-Net using a subvolume of one SBF-SEM dataset in 3dfullres mode. The resulting model is then used to infer the location of ER and mitochondria in the whole vol-ume of multiple datasets. (ii) An image analysis pipeline (see Methods) detects the ER-mitochondria contact areas that are equal or less than the search

distance (10–50 nm) from the nearest mitochondrion. (iii) The contacts may be visualized in 3D: orange contact is shown at a highlighted mitochondrion (arrow), ER is represented by green contour lines for clarity. (iv) Contacts detected can be mapped back to the original data for verification. A random selection of contacts from the 50 nm search distance collection are shown. **(C)** Plots of the total ER-mitochondria contact surface area per mitochondrion versus the surface area of the mitochondrion, for each search distance (indicated in gray box, nm). (i) scatter plots, $n_{mito}$ = 1,494–2,746 (control), 1,432–2,421 (rapamycin); 10–50 nm. (ii) contour plots are shown overlaid to compare the distribution between control and rapamycin. **(D)** Mean ER-mitochondria contact surface area per cell for each search distance. Each cell is represented as a dot, the mean ± sd is shown by a crossbar; $n_{cell}$ = 3 (control), 4 (rapamycin); $P$-values from Student's $t$ test with Welch's correction. The individual values for panels C and D are available at https://doi.org/10.5281/zenodo.15582238.

supported these observations (Figs 5B and S9). This suggested that LaBeRling of discrete ER-endosome or ER-lysosome contacts was possible, while Sec61β relocalization engulfed the target organelle and distorted any pre-existing ER-MCSs. The similar results at lipid droplets likely reflects that ER-lipid droplet contacts are more extensive than the discrete contacts at other organelles [21]. Given that the relocalization of LBR-FKBP-GFP can be used generically to mark different types of ER-MCS, we suggest that LaBeRling can be used as a multipurpose label for ER-MCSs.

## LBR sterol reductase activity is not required for ER-MCS labeling

We wondered if LBR was unique in being able to be used in this way to label ER-MCSs. To look at this, we investigated the relocalization of three other proteins: emerin, LAP2β, and BAF; which each localize at least partially to the inner nuclear envelope. Each protein was tagged at the N-terminus with FKBP-GFP and expressed in HCT116 cells alone (control), or coexpressed with either Stargazin-mCherry-FRB or MitoTrap, and all cells were treated with rapamycin (200 nM). We found that each of the relocalized proteins completely coated the surface of the PM or mitochondria in interphase or mitosis (S10 Fig). This suggests that something unique to LBR meant that it can be used as an inducible ER-MCS marker.

LBR transmembrane (TM) regions are important for sterol reductase activity, essential in the cholesterol biosynthesis pathway [29]. To determine if the sterol reductase function is required to label MCSs, we generated two cholesterol synthesis point mutants (N547D or R583Q) associated with disease and tested their ability to inducibly label ER-PM MCSs in mitosis [30]. Both mutants were relocalized to discrete puncta that were indistinguishable from those formed after relocalization of the WT LBR-FKBP-GFP to Stargazin-mCherry-FRB (Fig 6A and 6B). We made a series of truncation constructs to try to isolate the minimal region needed for inducible labeling of ER-MCSs. The two smallest constructs of this series contained either the first two TM domains or the first TM domain only (GFP-FKBP-LBR(1-288) or FKBP-LBR(1-245)-GFP, respectively). Surprisingly, both of these constructs formed discrete clusters upon relocalization, similar to the full-length protein (GFP-FKBP-LBR) (Fig 6B and 6C). These results suggest that the property of MCS targeting is contained in the N-terminal region and first TM domain; however, further truncations of this region resulted in mislocalization of the construct from the ER. Like the point mutants, the truncated constructs are not predicted to have any sterol reductase activity, so we can rule out cholesterol biosynthesis as the reason why LBR can be used to label ER-MCSs.

Rather than a molecular determinant in the N-terminus of LBR, an alternative possibility is that expression level of LBR is such that the density of protein in the ER is sufficient to label contact sites, yet is not high enough to distort them. Overexpression of LBR-FKBP-GFP results in similar LaBeRling of contact sites to endogenously tagged LBR (for example, compare Figs 1 and 2), so distortion of contact sites such as that seen with Sec61β and other proteins, is not possible to mimic with higher LBR expression. Therefore, we sought to reduce the expression of FKBP-GFP-Sec61β to see if labeling of contact sites was possible. Expression of FKBP-GFP-Sec61β under a PGK or crCMV promoter dramatically reduced the fluorescence of FKBP-GFP-Sec61β in the ER, to levels similar to endogenously tagged LBR (S12A and S12D Fig). Similarly, overexpression of LBR-FKBP-GFP under a CMV promoter resulted in ER fluorescence approaching that of FKBP-GFP-Sec61β under the same promoter (S12A and S12D Fig). Even with

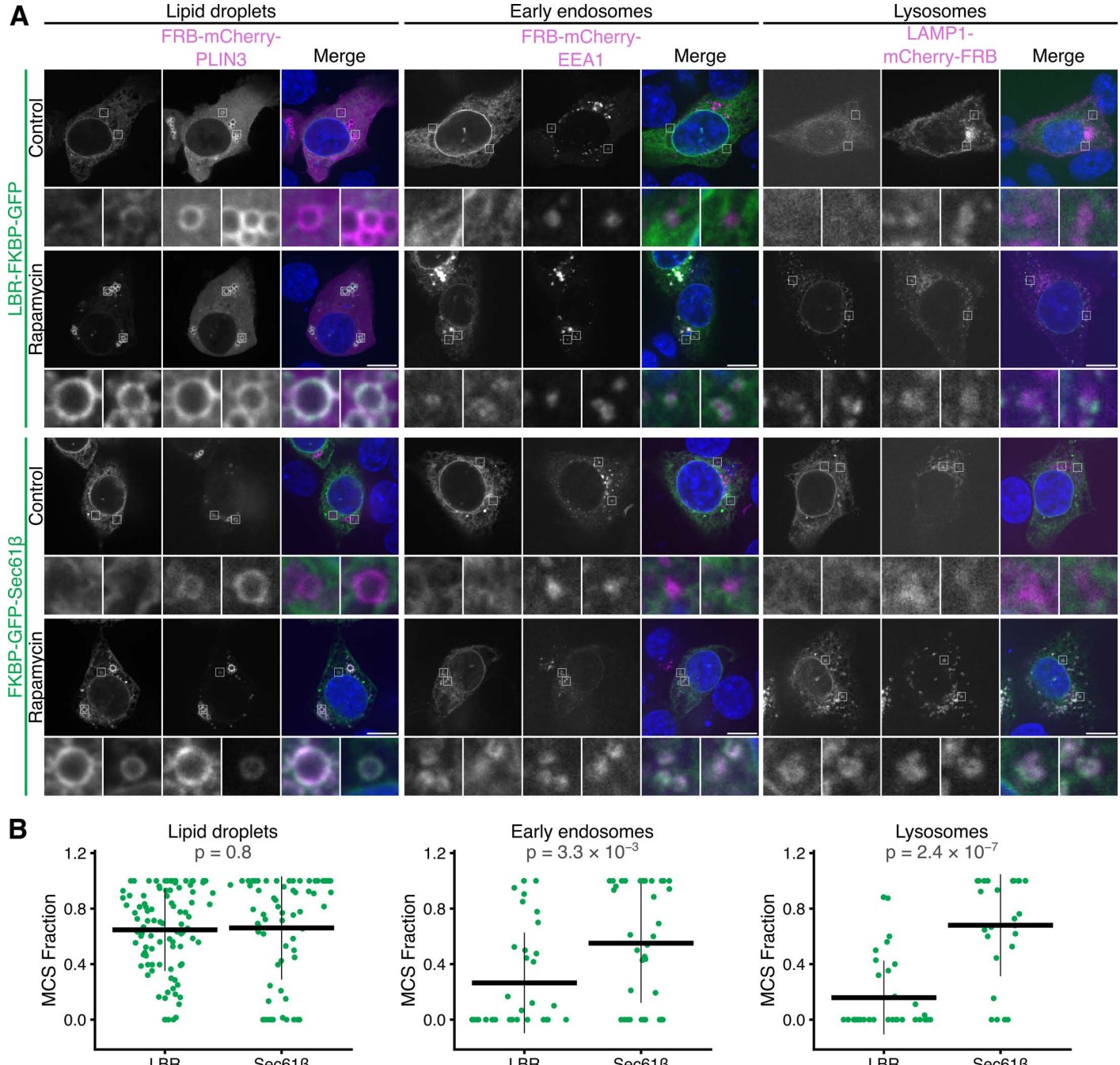

**Fig 5. Using LBR-FKBP-GFP relocalization to mark ER-lipid droplet, ER-endosome, or ER-lysosome contact sites. (A)** Example micrographs of HCT116 cells expressing either LBR-FKBP-GFP or FKBP-GFP-Sec61β (green) together with the indicated mCherry-FRB tagged protein anchor local- izing at the target membrane (magenta), either lipid droplets (FRB-mCherry-PLIN3), early endosomes (FRB-mCherry-EEA1), or lysosomes (LAMP1-mCherry-FRB), and stained with DAPI (blue). Lipid droplet number was increased by incubation with oleic acid (200 μM) for 17 h. Similar treatment reported to make no significant change to ER-lipid droplet contacts [11]. Relocalized samples were treated with rapamycin (200 nM) for 30 min before fixation. Control samples were not treated with rapamycin. A single slice from a z-stack of an interphase or metaphase cell is shown. Scale bars, 10 μm; zooms are 7.3× expansion of the ROI. **(B)** Plot of the fraction of the organelle perimeter that is above the threshold. Quantification is of line profiles like those shown in S9 Fig. *P*-values, Tukey's HSD *post-hoc* test; $n_{cell}$ = 4–29. The individual values for panel B are available at https://doi.org/10.5281/zenodo.15582238.

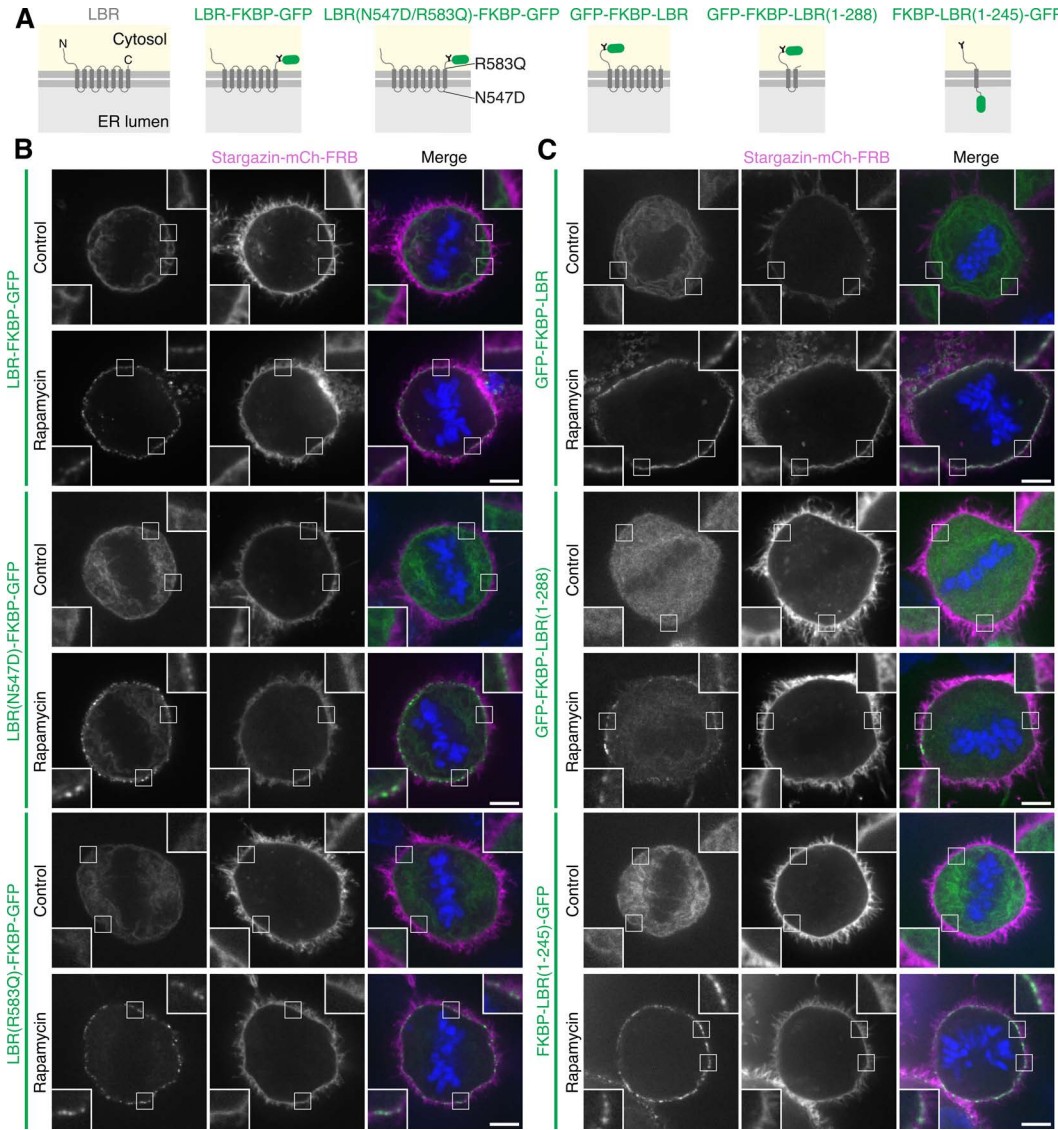

**Fig 6. Truncated LBR has similar labeling to the full-length protein. (A)** Schematic of GFP- and FKBP-tagged LBR, sterol reductase point mutants (N547D or R583Q) and truncated proteins (expressing LBR amino acids 1–288 or 1–245). Example micrographs of HCT116 cells expressing FKBP- and GFP-tagged LBR, LBR sterol reductase mutants **(B)** or truncated LBR **(C)** (green) alongside Stargazin-mCherry-FRB (magenta), stained with DAPI (blue). Relocalized samples were treated with rapamycin (200 nM) for 30 min before fixation. Control samples were not treated with rapamycin. Scale bars, 5 µm; insets, 2.5× expansion of ROI.

lower expression, relocalization of FKBP-GFP-Sec61β to Stargazin-mCherry-FRB resulted in large contacts that were much larger than those produced by LaBeRling or by contacts marked by mScarlet-I3-6DG5-MAPPER in the same cell (S12B and S12C Fig). Quantification of the fluorescent clusters after relocalization showed that the median cluster area and the density of clusters of LBR-FKBP-GFP was significantly lower than those seen with FKBP-GFP-Sec61β even at the lowest expression levels (S12E and S12F Fig). These experiments suggest that simple expression differences are not sufficient to explain the difference in activity, and argue that LBR is especially suited for contact site labeling.

## Using LaBeRling to investigate novel contact sites: ER-Golgi contact sites in mitosis

Contact sites between the ER and *trans*-Golgi network (TGN) have been described [14]. The Golgi undergoes massive remodeling during mitosis [3], but due to the lack of inducible labeling techniques, the status of ER-Golgi MCSs during mitosis has not been studied. Having established LaBeRling, we sought to establish whether we could label ER-Golgi MCSs and if so, to test if they persist during mitosis. HTC116 cells transiently co-expressing LBR-FKBP-GFP and FRB-mCherry-Giantin(3,131−3,259) were imaged following relocalization with rapamycin (200 nM) (Fig 7A). We compared these to similarly prepared samples co-expressing FKBP-GFP-Sec61β and FRB-mCherry-Giantin(3,131−3,259). In interphase cells, LBR-FKBP-GFP formed distinct puncta when relocalized to FRB-mCherry-Giantin(3,131−3,259) Golgi structures. By contrast, FKBP-GFP-Sec61β coated much of the interphase Golgi structures after relocalization (Fig 7A and 7B). Similar LaBeRling was also observed in RPE-1, Cos-7, and HeLa cells (S6 Fig). These results suggest that LBR can be used to selectively label ER-Golgi MCSs.

In mitotic cells, relocalized LBR-FKBP-GFP was observed to be at puncta coinciding with Golgi fragments at prometaphase and at metaphase (Fig 7A). This relocalization pattern could be observed in live mitotic cells (S10 Video) forming with dynamics similar to that of other ER-MCS labeling events. Moreover, in mitotic HCT116 LBR-FKBP-GFP knock-in cells the same labeling was present confirming that the labeling of ER-Golgi MCSs in mitosis was not due to overexpression of the LBR protein (Fig 8A). Again, by contrast, large patches of FKBP-GFP-Sec61β signal were observed at relocalized at FRB-mCherry-Giantin(3131-3259) Golgi fragments in prometaphase and metaphase cells (Fig 7A and 7B). In fact, the Golgi haze appeared to be cleared from the cytoplasm and that Golgi was recruited to the ER. These results suggest that selective labeling of ER-Golgi MCSs by LBR was possible and was distinguishable from nonspecific heterodimerization between ER and Golgi.

The persistence of LaBeRling in mitotic cells suggests that ER-Golgi MCSs are maintained during mitosis. To examine these contacts in further detail, we examined ER-MCSs in mitotic HCT116 LBR-FKBP-GFP knock-in cells by SBF-SEM (Figs 8B and S13). Small clusters of vesicles were readily observable within 30 nm of ER in metaphase and telophase cells. These clusters match the mitotic Golgi clusters described in EM images of HeLa cells in prometaphase, metaphase, and telophase [31], and more recently in NRK cells in prophase, metaphase, and late anaphase [32]. Together, these data suggest that ER-Golgi MCSs are maintained in mitotic cells and can be labeled by the relocalization of LBR-FKBP-GFP to Golgi membranes using FRB-mCherry-Giantin(3131-3259).

## Discussion

In this study, we developed a new method, LaBeRling, for inducible labeling of ER-MCSs using the ER protein LBR. Unlike previous inducible labeling approaches, LaBeRling does not alter existing ER-PM or ER-mitochondria MCSs and it does not induce artificial contacts. Moreover, labeling is fast (<2 min) and persists over many minutes without distortion of MCSs, so it is ideal for labeling MCSs at discrete stages of the cell cycle. Finally, we used this method to demonstrate the presence of ER-Golgi MCSs in mitosis, at a time of mitotic Golgi dispersal.

LaBeRling uses heterodimerization of LBR with a generic anchor protein on the target membrane. This anchor protein does not need to localize to MCSs, and can be readily interchanged, so that LBR can be used to label a range of ER-MCSs (between PM, mitochondria, early endosomes, lysosomes, lipid droplets, and Golgi) in interphase and mitotic cells. In the past, other approaches have used MCS tethering proteins for inducible labeling [33,34], with the logic that this will increase specificity. This is problematic because both the overexpression of tethering proteins themselves and their subsequent heterodimerization can distort MCSs and induce artificial contacts [33,35], as similarly seen using membrane targeting anchors [17,22,23]. LaBeRling delivers more universal labeling of contact sites since the initial coverage of the target membrane and the ER is diffuse and homogeneous yet specific to the respective compartments. We envisage that LaBeRling could also be applied to label other ER-membrane contacts, for example, ER-peroxisome and ER-autophagosome MCSs.

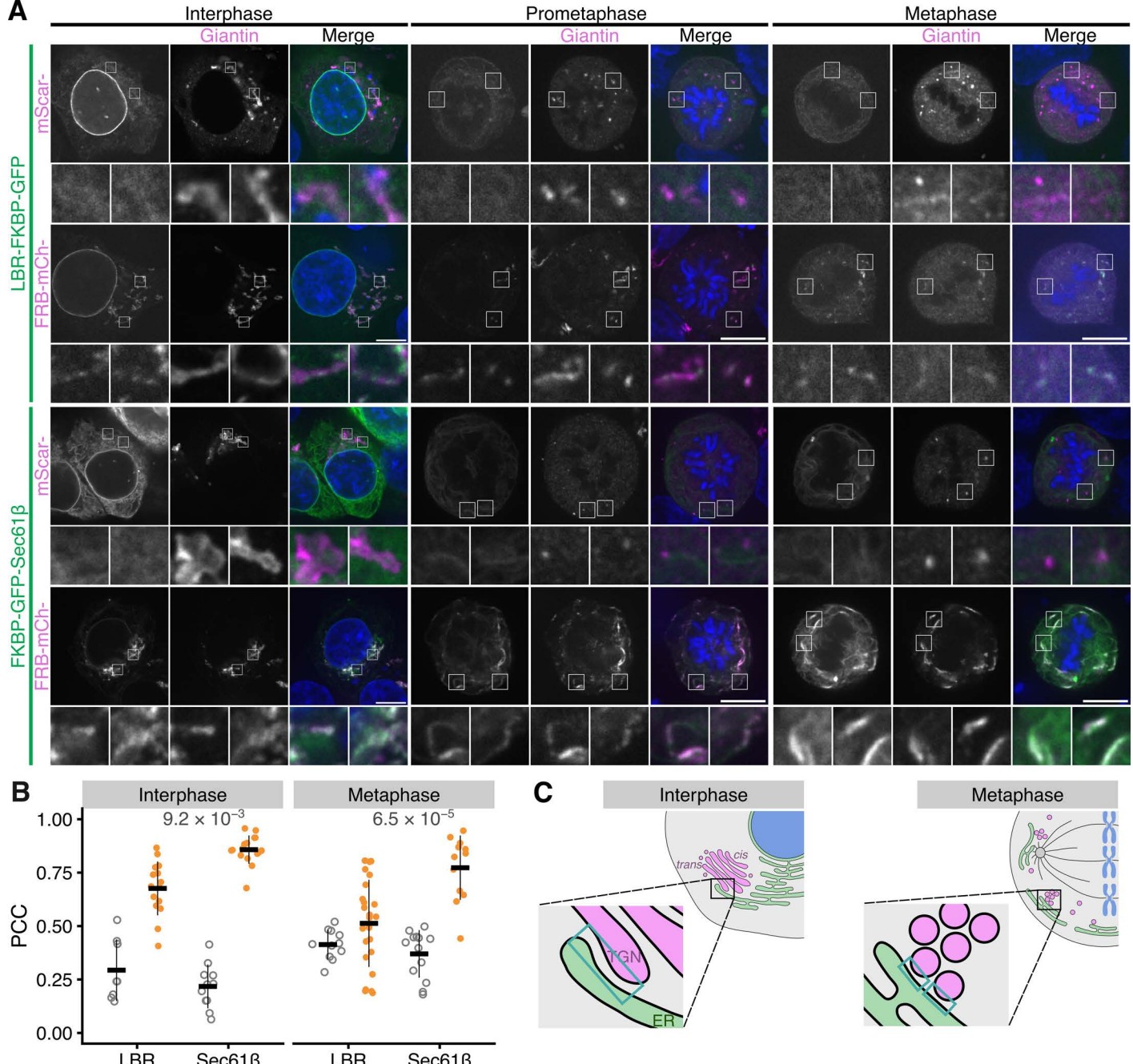

**Fig 7. Using LBR-FKBP-GFP relocalization to identify novel ER contact sites. (A)** Example micrographs of synchronized HCT116 cells expressing either LBR-FKBP-GFP or FKBP-GFP-Sec61β (green) together with mScarlet-Giantin or FRB-mCherry-Giantin(3131-3259) (magenta), and stained with DAPI (blue). All samples were treated with rapamycin (200 nM) for 30 min before fixation. Single slices from z-stacks of an interphase, early mitotic or metaphase cell are shown. Scale bars, 10 μm; zooms, 5.8× expansion of ROI for interphase cells and 4× expansion for prometaphase and metaphase. **(B)** Plots of Pearson's correlation coefficient between LBR-FKBP-GFP or FKBP-GFP-Sec61β and either mScarlet-Giantin (gray) or FRB-mCherry-Giantin(3,131-3,259) (orange) at the indicated cell cycle stage. Markers, cells. Bars indicate mean ± sd. P-values, Tukey's HSD *post-hoc* test; $n_{cell}$ = 8–24. **(C)** Schematic representation of ER-Golgi contacts in interphase or metaphase cells, with example contact sites indicated in the expanded region (blue box). The individual values for panel B are available at https://doi.org/10.5281/zenodo.15582238.

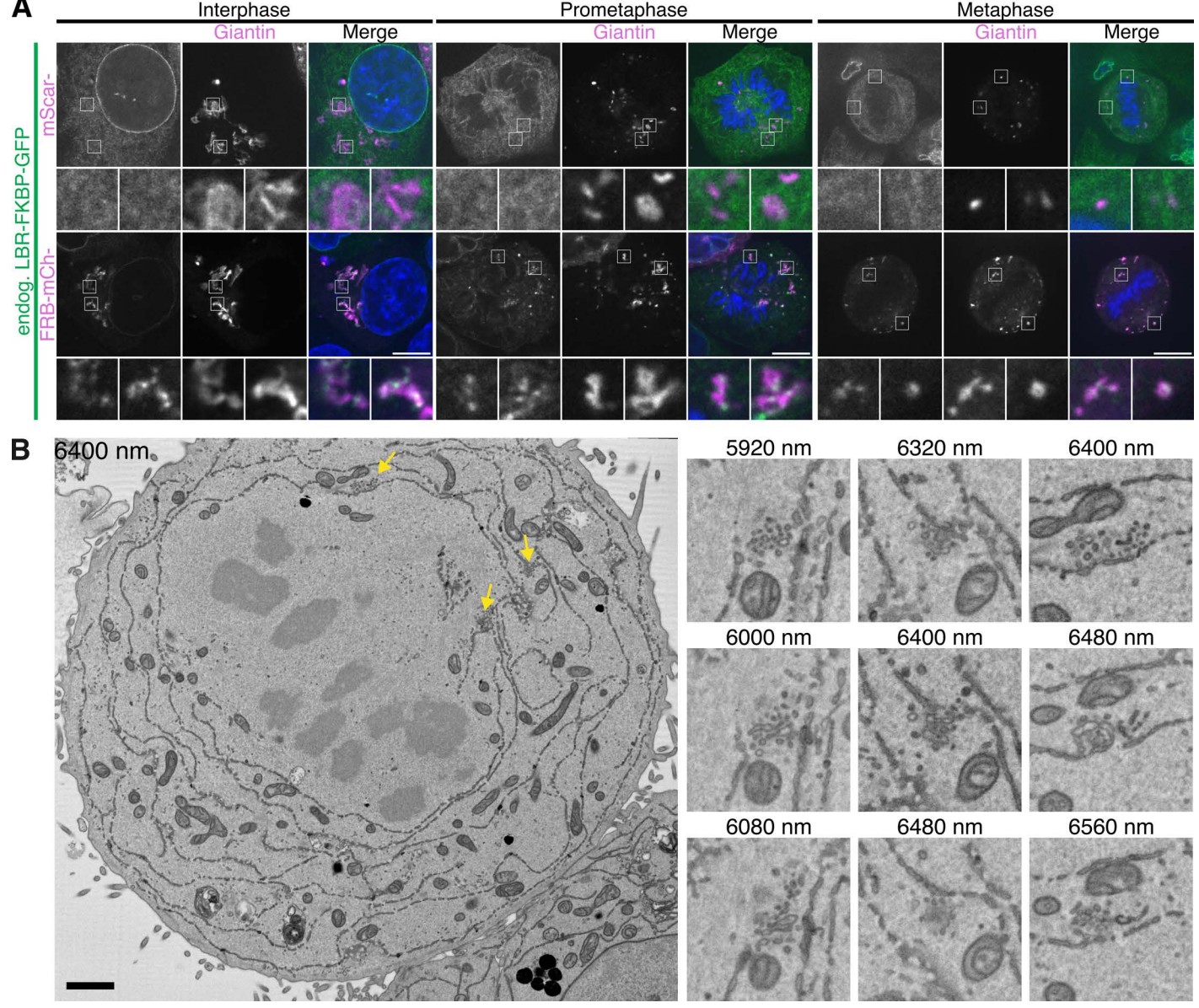

**Fig 8. LaBeRling ER-Golgi membrane contact sites in mitosis. (A)** Example micrographs of synchronized HCT116 LBR-FKBP-GFP knock-in cells expressing either mScarlet-Giantin or FRB-mCherry-Giantin(3131-3259) (magenta), and stained with DAPI (blue). All samples were treated with rapamycin (200 nM) for 30 min before fixation. Single slices from z-stacks of an interphase, early mitotic or metaphase cell are shown. Scale bars, 10 µm; insets, 4× expansion of ROI. **(B)** Single slices from SBF-SEM dataset of a metaphase HCT116 LBR-FKBP-GFP knock-in cell transiently expressing Mitotrap and treated with rapamycin (200 nM) are shown. Depth of each slice within the dataset (nm) is indicated. Example Golgi clusters are shown by yellow arrows on the full slice image. Three sequential slices of these regions (3× expansion) are shown beside. Scale bars, 2 and 0.5 µm on zoom region.

Our serendipitous discovery that LBR can be used to label MCSs is intriguing: what makes LBR so special? We saw that the anchor protein remains homogeneously distributed after LBR relocalization and so the MCS specificity of the labeling seems to be driven by LBR, from the ER side. Why LBR behaves in this way, whereas other ER proteins do not, is unresolved. Cholesterol synthesis activity was not essential for LaBeRling because point mutants and truncated LBR proteins that lack sterol reductase activity all labeled EM-PM MCSs similarly to full-length WT LBR. It is possible that the intermembrane distance at the ER-MCSs

is such that the size of LBR-FKBP-GFP is in a "sweet spot" to be immobilized by the heterodimerization procedure, but not so large as to crosslink membranes outside of the MCS. In support of this possibility, there is evidence that altering the spacing in optogenetic heterodimerization of ER-PM linkages affects labeling efficiency [35]. Adjusting linkage distance of ER-PM contacts has also been indicated to affect protein translocation into the MCS [18,25]. However, the intermembrane distances of MCSs are reported to be rather variable, and we observed that various anchor proteins can be used successfully, which argues against the idea that physical spacing can explain the labeling behavior. A further possibility is that the density of LBR in the ER is lower than other proteins so that its relocalization does not distort contacts. However, we saw no distortion at higher expression levels, and also that reducing the expression of other ER proteins did not allow for specific labeling, which argues against this possibility. Recent work points to LBR having affinity for the $IP_3$ receptor, which could potentially provide a molecular explanation for why it can be used to inducibly label contacts [36]. Whatever the mechanism, the observation that sterol reductase activity is not required means that the LBR point mutants can be used to ensure the local sterol environment at the MCS is not modified by labeling. This may be important because MCSs are reported to be enriched with cholesterol [37–39], and an ideal labeling method would not perturb the lipids at the endogenous MCS. As a corollary to this, the report of LBR stabilizing ER-MCSs during mitosis by connecting the $IP_3$ receptor in the ER with VDAC2 on mitochondria, thereby regulating calcium flux and energy levels [36], brings a note of caution. LaBeRling mitochondria-ER MCSs during mitosis, may prime sites for complex formation and affect the calcium flux. The formation of this complex is regulated by phosphorylation, and so this issue is unlikely to affect LaBeRling in nondividing cells nor is it predicted to affect LaBeRling of ER contacts with other organelles. In interphase, ER-TGN MCSs have been described [2,14], and we used a generic Golgi anchor protein and LBR-FKBP-GFP to detect ER-Golgi MCSs, which were presumably ER-TGN contacts. We used our inducible method to show that ER-Golgi MCSs are maintained in mitosis, a time when the Golgi has been disassembled [3]. To our knowledge this is the first labeling of these MCSs during mitosis but is corroborated by evidence from EM studies. Briefly, the Golgi is disassembled by severing the Golgi ribbons into stacks and then dispersing the stacks into Golgi "blobs" and "haze" [40], which correspond to vesicular clusters and single vesicles, respectively. Each Golgi cluster likely contains a mixture of vesicles from TGN, *cis-* and medial-Golgi. For example, a subset of the vesicles within the mitotic Golgi clusters had TGN markers in HeLa cells by EM [31], and an incomplete overlap of different Golgi markers stained within mitotic Golgi clusters was detected by light microscopy in NRK cells [41]. Therefore, ER-TGN contacts are an efficient way for the ER to remain in contact with Golgi clusters. The function of ER-Golgi contacts maintained during mitosis is unclear. The spindle operates in an "exclusion zone" which is largely membrane free [42]. Maintained ER-Golgi contacts could serve simply to exclude clusters from the spindle area and to prevent these membrane fragments from interfering with chromosome segregation. Another possibility is that the ER-Golgi contacts may coordinate reassembly by allowing the clusters to surf on the ER toward the spindle pole and the midbody where the Golgi twins coalesce and begin to reassemble [43].

We used LaBeRling to probe other ER-MCSs in mitosis. For example, we saw that the total number of ER-PM MCSs was reduced at metaphase compared with interphase cells. This observation is seemingly at odds with our observation that the density of contacts remained constant. However, our density measurement uses a surface constructed from the contact sites rather than the PM surface area for normalization. So while the net number of contacts decreases, the spacing between them is maintained as the PM is drawn up and away from the underlying ER [44]. How this process is regulated and how MCS are modified during mitosis are interesting questions for the future.

To conclude, the method we describe to label ER-MCS using LBR can be applied to a wide variety of questions at many different types of contact in cells. Options for further development include modifying the mode of heterodimerization to allow for reversible and repeatable labeling, and exploiting the system to deliver molecules to MCSs to tweak their properties.

## Methods

### Molecular biology

The following plasmids were available from Addgene or previous work: pEBFP2-N1 (Addgene #54595); EGFP-BAF (Addgene #101772); Emerin pEGFP-C1 (637) (Addgene #61993); FKBP-alpha(740-977)-GFP (Addgene #100731);

FKBP-GFP-Sec61β (Addgene #172442); FRB-mCherry-Giantin (Addgene #186575); GFP-EEA1 (Addgene #42307); LAMP1-mCherry-FRB (Addgene #186576); LAP2 Full I pAcGFP-N1 monomeric GFP (1317) (Addgene #62044); LBR pEGFP-N2 (646) (Addgene #61996); pFKBP-GFP-C1 [45]; pMaCTag-P05 (Addgene #120016); pMito-mCherry-FRB (Addgene #59352); pMito-mCherry-FRB(T2098L); pMito-dCherry-FRB (Addgene #186573); pmScarlet-Giantin-C1 (Addgene #85048); SH4-FRB-mRFP (Addgene #100741); Stargazin-dCherry-FRB (Addgene #172444); Stargazin-GFP-LOVpep (Addgene #80406); Stargazin-mCherry-FRB (Addgene #172443) [26,46–48].

To generate the pFRB-mCherry-C1 vector, FRB was amplified from pMito-mCherry-FRB plasmid (using CGCGGCT AGCGGCCACCATGATCCTCTGGCATGAGATGTGGCATGAAGGC and TCGCaccggtggGCCGGCCTGCTTTGAG ATTC-GTCGGAACAC) and inserted into pmCherry-C1 vector (using NheI-AgeI). pmCherry-C1 vector was made by substituting mCherry for EGFP in pEGFP-C1 [Clontech] by AgeI-XhoI digestion.

LBR-FKBP-GFP was generated by cutting LBR from LBR-mCherry using BamHI-KpnI sites and ligating into pFKBP-GFP-N1 plasmid, where FKBP was inserted into pEGFP-N1 at BamHI-AgeI sites. Similarly, FKBP-GFP-emerin was made by digestion of Emerin pEGFP-C1 (637) and ligation into FKBP-GFP-C1 plasmid using XhoI and BamHI sites. To clone FKBP-GFP-LAP2β, BglII-SalI sites were introduced at either end and a C-terminal stop codon to LAP2β amplified by PCR from LAP2 Full I pAcGFP-N1 monomeric GFP (1317) plasmid template (primers aagcttAGATCTATGCCGGAGTTCCTAGAGG and tcgagGTCGACCTAg-CAGTTGGATATTTTAGTATCTT GAAG), and ligating into pFKBP-GFP-C1. FKBP-GFP-BAF was generated by digestion of mCherry-BAF plasmid and ligation into pFKBP-GFP-C1 at BglII and Acc65I sites. The mCherry-BAF plasmid used in this clon-ing was made by PCR amplification of the BAF-encoding region from EGFP-BAF (primers aagcttAGATCTATGACAACCTCCCA AAAGC and tcgagAAGCTTCTACAAGAAGGCATCACACC) and ligation into pmCherry-C1 vector (BglII-HindIII). For expression of LBR-FKBP-GFP under different promoters the CMV promoter was exchanged for PGK or crCMV (available from previous work).

LBR point mutations (N547D or R583Q) were introduced by site-directed mutagenesis using LBR-FKBP-GFP plasmid template. Primer with mismatches were designed to introduce the mutation N547D (GTTCGCCACCCC GATTACTTG-GGTGATCTCATC and GATGAGATCACCCAAGTAATCGGGGTGGCGAAC) or R583Q (CATGTTGC TTGTCCAC-CAAGAAGCTCGTGACG and CGTCACGAGCTTCTTGGTGGACAAGCAACATG).

The LBR truncations were generated by PCR amplication of the LBR region from LBR pEGFP-N2 (646) template. To generate the GFP-FKBP-LBR(1-288) plasmid, the primer set aagacaGAATTCaATGCCAAGTAGGAAATTTGCC G and tcgagGGATCCTTAATCAATAAGAGGCGTTCCTTCTACAAC was used. The PCR product was ligated into pEGFP-FKBP-C1 using EcoRI-BamHI sites. FKBP-LBR(1-245)-GFP truncation was made by amplifying the region encoding amino acids 1-245 (using aagacaGGTACCGTGAACCGTCAGATCCGCTAG and tcgagACCGGTagAGG AGG-GAAATTCAGAAGACTGGGATCTTTC). The PCR product was ligated in substitution of the AP2A1 encoding region in FKBP-alpha(740-977)-GFP (using KpnI-AgeI).

The PM anchor Stargazin-EBFP2-FRB was made by PCR of Stargazin encoding region from Stargazin-GFP-LOVpep (primers gcggctagcATGGGGCTGTTTGATCGAGGTGTTCA and TTTACTCATGGATCCt tTACGGGCGTGGTCCGG) and then ligating into pEBFP2-N1. Stargazin-EBFP2 was amplified from the resulting plasmid (primers aagcttGCTAGCcATG-GGGCTGTTTGATCGAGG and tcgagGGTACCccCTTGTACAGCTCGTCCATGC), excluding the stop codon, and ligated N-terminal to FRB in a plasmid encoding Stargazin-dCherry-FRB. SH4-FRB-EBFP2 was made by substituting mRFP of SH4-FRB-mRFP with EBFP2 from pEBFP2-N1 (using AgeI-NotI).

Endosomal anchor FRB-mCherry-EEA1 was made by cutting full-length EEA1 from GFP-EEA1 using XhoI-BamHI and ligating into pFRB-mCherry-C1. Lipid droplet anchor FRB-mCherry-PLIN3 was made from a mGFP-PLIN3 full-length plasmid that was generated by DNA synthesis and Gibson Assembly, cutting FRB-mCherry from FRB-mCherry plasmid (NheI-BsrGI) and substituting for the mGFP of mGFP-PLIN3.

PM anchor binding to rapalog, Stargazin-mCherry-FRB(T2098L), was made by cutting the Stargazin-encoding sequence from Stargazin-mCherry-FRB plasmid (NheI-BamHI) and paste in place of the mitochondrial targeting sequence in pMito-mCherry-FRB(T2098L).

The template plasmid for the C-terminal PCR tagging CRISPR method [49] encoding FKBP-GFP tag (pMaCTag-P05-FKBP-GFP) was generated by amplifying the region encoding FKBP-GFP from pFKBP-GFP-N1 and introducing BamHI-SpeI restriction sites (primers aagcttGGATCCCCGCCACCAATGGGAGTGCAGGTGG and tcgagACTAGTTTACTTGTACAGCTCGTCCATGCCGAGAGT), and then ligating in place of the GFP tag in available pMaCTag-P05 to give pMaCTag-P05 FKBP-GFP. To generate the PCR product for editing, LBR-specific tagging oligos were designed using the online design tool (M1, CGTGACGAGTACCACTGTAAGAAGAAATACGGCGTG GCTTGG-GAAAAGTACTGTCAGCGTGTGCCCTACCGTATATTTCCATACATCTACTCAGGTGGAGGAGGTAGTG and M2, TTTG-CAAATGGCAGCTGGAATTGCAGGAGTATTTTGTAGAAAAGCCAGAAGAGCAAAAAAAAGAGCA TTAGTAGAT GTATGATCTACACTTAGTAGAAATTAGCTAGCTGCATCGGTACC). Primers for genotyping PCR to confirm LBR-FKBP-GFP knock-in were AGAATTTGGGGGAAAGCAGG and CATCCTTACTTGTATTTTTCCTATG TTAACTG or AAGA-CAATAGCAGGCATGCT and CAGTGGCACCATAGGCATAA.

The mScarlet-I3-6DG5-MAPPER plasmid was designed based on the available GFP-MAPPER sequence and generated by DNA synthesis [18]. Briefly, the GFP-encoding sequence of GFP-MAPPER was replaced with mScarlet-I3 and the unwanted FRB sequence was substituted with a codon-optimized sequence of Neoleukin-2/15 (PDB code, 6DG5), which is structurally similar and close in mass to FRB. Linker sequence lengths were maintained similar to that in the GFP-MAPPER sequence.

## Cell biology

HCT116 (CCL-247; ATCC) cells and lines derived from HCT116 were maintained in DMEM supplemented with 10% FBS and 100 U mL$^{-1}$ penicillin/streptomycin. All cell lines were kept in a humidified incubator at 37 °C and 5% CO$_2$. Cells were routinely tested for mycoplasma contamination by a PCR-based method.

HCT116 LBR-FKBP-GFP CRISPR knock-in cells were generated using a C-terminal PCR tagging CRISPR method [49]. HCT116 cells were transfected with the M1-M2 PCR cassette and a plasmid encoding Cas12a (pVE13300). Cells were selected and maintained in media supplemented with puromycin dihydrochloride (Gibco) at 1.84 μM. Populations of cells positive for GFP signal were selected by FACS and the positive pools of cells were characterized by genotyping PCR, western blot and microscopy.

For transient transfection, Fugene-HD (Promega) was used to transfect HCT116 and RPE-1 cells, and GeneJuice (Merck) used for HeLa and Cos-7 cells, each according to the manufacturer's instructions. A total of 1 μg DNA with 3 μL reagent was used per fluorodish or well of a six-well plate for each transfection reagent. With the exception of LBR-FKBP-GFP expressed endogenously, the expression of all constructs was via transient transfection.

Heterodimerization of FKBP and FRB tags was induced through addition of rapamycin to media at a final concentration of 200 nM. For fixed cell experiments, 200 nM rapamycin (J62473, Alfa Aesar) was prepared in complete media 2 mL per well of a 6-well plate; growth media was removed, rapamycin-containing media added, and then the plate was returned to the incubator until fixation. To apply rapamycin to live cells on the microscope, rapamycin solution (400 nM) in imaging media (Leibovitz's L-15 Medium, no phenol red, supplemented with 10% FBS) was diluted (1:2) to final concentration 200 nM using media in the dish. Similarly, heterodimerization of FKBP and FRB(T2098L) tags was induced with rapalog (A/C Heterodimerizer, Takara, 635057) at a final concentration of 5 μM, applied to cells similarly to the rapamycin treatment described above. To visualize DNA in live cells, dishes were incubated for 15–30 min with 0.1 μM SiR-DNA (Spirochrome) prepared in complete media. Cells were selected for moderate expression of the anchor construct, to minimize the possibility that organelle morphology, and MCSs were affected by each anchor.

Cells were synchronized by treatment with thymidine (2 mM, 16 h), followed by washout, 7–8 h incubation and then RO-3306 treatment (9 μM, 16 h). Cells were released from synchronization and incubated for 15 min or 30 min before applying rapamycin solution (final concentration 200 nM) and incubating a further 30 min before fixation.

To increase the number of lipid droplets, the media on cells were supplemented with oleic acid (O3008, Sigma) at 200 µM) for around 17 h before fixation.

Thapsigargin treatment was used to induce an increase in ER-PM contacts, as described [18,50]. Thapsigargin (T9033, Sigma) solution was prepared at 1.5 mM in DMSO and applied to cells at 1 µM final concentration in imaging media.

## Fluorescence methods

For fixed-cell imaging, cells were seeded onto glass cover slips (16 mm diameter and thickness number 1.5, 0.16–0.19 mm). Cells were fixed using 3% PFA/4% sucrose in PBS for 15 min. After fixation, cells were washed with PBS and then incubated in permeabilization buffer (0.5% v/v Triton X-100 in 1 × PBS) for 10 min. Cells were washed twice with PBS, before 45–60 min blocking (3% BSA, 5% goat serum in 1 × PBS). Antibody dilutions were prepared in blocking solution. After blocking, cells were incubated for 1 h with primary antibody, PBS washed (three washes, 5 min each), 1 h secondary antibody incubation, PBS washed (three washes, 5 min each), mounted with Vectashield containing DAPI (Vector Labs ) and then sealed. PhenoVue Fluor 568—Concanavalin A dye (CP95681) stain was added to coverslips at 50 µg mL$^{-1}$ in HBSS for 15 min at room temperature, followed by three 5 min PBS washes before mounting and sealing as described above.

## Western blotting

For Western blotting, cells were harvested, and lysates were prepared by sonication of cells in extraction buffer (8 M urea, 50 mM Tris, and 150 mM 2-mercaptoethanol) for HCT116 LBR-FKBP-GFP knock-in cell characterization. All lysates were incubated on ice for 30 min, clarified in a benchtop centrifuge (20,800*g*) for 15 min at 4 °C, boiled in Laemmli buffer for 10 min, and resolved on a precast 4%–15% polyacrylamide gel (Bio-Rad). Proteins were transferred to nitrocellulose using a Trans-Blot Turbo Transfer System (Bio-Rad). The following antibodies were used: rat monoclonal anti-GFP (3h9, Chromotek) at 1:1000 in 3% BSA TBST; mouse anti-LBR antibody (polyclonal) (SAB1400151, Sigma-Aldrich) at 1:500 in 2% milk TBST; goat anti-rat IgG-Peroxidase antibody (A9037, Sigma) at 1:5000 in 5% milk TBST; sheep anti-mouse IgG-HRP (NXA931, Cytiva) at 1:5000 in 5% milk TBST; donkey anti-rabbit IgG-HRP (NA934V, Cytiva) at 1:2500 in 2% milk TBST; loading control HRP-conjugated mouse anti-β-actin (C4) (sc-47778, Santa Cruz Biotechnology) at 1:20,000 in 2% milk TBST.

## Microscopy

As described previously, images were captured using a Nikon CSU-W1 spinning disc confocal system with SoRa upgrade (Yokogawa) with either a Nikon, 100×, 1.49 NA, oil, CFI SR HP Apo TIRF or a 63×, 1.40 NA, oil, CFI Plan Apo objective (Nikon) with optional 2.8× intermediate magnification and 95B Prime camera (Photometrics) [26]. The system has a CSU-W1 (Yokogawa) spinning disk unit with 50 µm and SoRa disks (SoRa disk used), Nikon Perfect Focus autofocus, Okolab microscope incubator, Nikon motorized xy stage and Nikon 200 µm z-piezo. Excitation was via 405, 488, 561 and 638 nm lasers with 405/488/561/640 nm dichroic and Blue, 446/60; Green, 525/50; Red, 600/52; FRed, 708/75 emission filters. Acquisition and image capture was via NiS Elements (Nikon). All microscopy data were stored in an OMERO database in native file formats.

## Serial block face-scanning electron microscopy

Preparation of samples for serial block face-scanning electron microscopy (SBF-SEM) was performed as described previously [26,51]. Briefly, HCT116 LBR-FKBP-GFP knock-in cells expressing MitoTrap were plated onto gridded dishes and prior to imaging, incubated for around 30 min with 0.5 µM SiR-DNA (Spirochrome) to visualize DNA. Using light microscopy, live cells were imaged to confirm the induced labeling following rapamycin (200 nM) treatment. Control

cells not treated with rapamycin were imaged in parallel and the coordinate position of the cell of interest recorded for correlation by SBF-SEM. Cells were washed twice with phosphate buffer (PB) before fixing (2.5% glutaraldehyde, 2% paraformaldehyde, 0.1% tannic acid (low molecular weight) in 0.1 M phosphate buffer, pH 7.4) for 1 h at room temperature. Samples were washed three times with PB and then post-fixed in 2% reduced osmium (equal volume of 4% $OsO_4$ prepared in water and 3% potassium ferrocyanide in 0.1 M PB solution) for 1 h at room temperature, followed by a further three washes with PB. Cells were then incubated for 5 min at room temperature in 1% (w/v) thiocarbohydrazide solution, followed by three PB washes. A second osmium staining step was then included, incubating cells in a 2% $OsO_4$ solution prepared in water for 30 min at room temperature, followed by three washes with PB. Cells were then incubated in 1% uranyl acetate solution at 4 °C overnight. This was followed by a further three washes with PB. Walton's lead aspartate was prepared adding 66 mg lead nitrate (TAAB) to 9 mL 0.03 M aspartic acid solution at pH 4.5, and then adjusting to final volume of 10 mL with 0.03 M aspartic acid solution and to pH 5.5 (pH adjustments with KOH). Cells were incubated in Walton's lead aspartate for 30 min at room temperature and then washed three times in PB. Samples were dehydrated in an ethanol dilution series (30%, 50%, 70%, 90%, and 100% ethanol, 5 min incubation in each solution) on ice, then incubated for a further 10 min in 100% ethanol at room temperature. Finally, samples were embedded in an agar resin (AGAR 100 R1140, Agar Scientific). SBF-SEM imaging was carried out by the Biomedical Electron Microscopy Unit at University of Liverpool, UK.

## Data analysis

Analysis of confocal z-stacks of LBR-FKBP-GFP clusters at the PM was by 3D Spot Finder in 3D Image Suite plugin in Fiji. Briefly, outputs were fed into R where the size and number of clusters was stored. Contact area was defined as half of the surface area of a cluster. The location of clusters was used to find the surface of the cell using alphashape3d. The total number of clusters divided by the surface area of the alpha shape was used to determine the density of clusters per cell. For analysis of cluster formation in movies of LBR-FKBP-GFP relocalization to the PM, a weka segmentation method was used in TrackMate/Fiji to define the clusters that formed and track individual clusters over time. The outputs of these TrackMate XML files were analyzed using TrackMateR [52]. Tracks shorter than 4 frames or those that terminated before 100 s were removed from analysis and the remainder analyzed for shape, intensity, and number over time.

Analysis of ER-PM contacts labeled by LBR-FKBP-GFP/MAPPER was done using a semi-automatated procedure to segment each channel in 3D using 3D spot finder in 3D Image Suite in Fiji. Output were processed in R, where the total numbers of clusters per cell was analyzed per cell. Comparisons between pre and post-rapamycin was done using paired t-tests on the individual cell data, with Holm–Bonferroni correction for multiple testing. Comparison between conditions was done using the experimental means using ANOVA with Tukey's post-hoc test.

Line profile data were harvested by manually outlining organelles of interest in the red channel, without sight of the green channel. Intensities for both channels along the profile were saved along with the background and maximum value for the image for normalization. Following import into R, the rle function was used to quantify segmented regions of the profile above a threshold.

For ER-mitochondria contact analysis, each SBF-SEM dataset was aligned using SIFT and cropped. A subvolume of one dataset (48 slices) was manually segmented for ER and mitochondria using IMOD. The segmentation and corresponding raw data were used to train nnU-Net v2 [53] running on a GPU workstation (Intel Core i9-7900X, 128 GB, with TITAN Xp GPU). The resulting model was used to infer the ER and mitochondria in all datasets. Visualization of the inference maps or the manual segmented IMOD models was done using ChimeraX. A series of Fiji/ImageJ scripts were used to process the output. Briefly, an exact Euclidean distance transform (EDT) was generated using the mitochondria channel to give a 3D volume of distances from each voxel to the nearest mitochondrion. Then the overlap between ER channel and EDT at distances of 10–50 nm was calculated in 10 nm increments, to give the ER regions (contacts) within the appropriate distance from the mitochondrion. This result was segmented in 3D to classify all of the contacts. In

addition, the mitochondria channel was also segmented in 3D. These outputs were processed in R to match each contact with its corresponding mitochondrion, which allowed the comparison of total contact surface area per mitochondrion with the mitochondrion surface area.

## Supporting information

**S1 Fig. Further controls for rapamycin treatment and expression of the plasma membrane anchor construct.** Example micrographs of HCT116 wild-type cells transiently expressing LBR-FKBP-GFP or FKBP-GFP-Sec61β (green) and HCT116 LBR-FKBP-GFP CRISPR knock-in cells, transiently expressing Stargazin-mCherry-FRB (magenta) as indicated and stained with DAPI (blue). Relocalized samples were treated with rapamycin (200 nM) for 30 min before fixation. Control samples were not treated with rapamycin. Scale bars, 10 µm; Insets, 3× expansion of ROI.
(TIFF)

**S2 Fig. HCT116 LBR-FKBP-GFP knock-in cells. (A)** Schematic diagram of C-terminal PCR tagging of LBR with FKBP-GFP. **(B)** FACS plots to show the collection of a mixed population of GFP-positive cells. **(C)** Agarose gel to show genotyping PCR results using indicated primers. **(D)** Western blot of lysates collected from parental cells or the edited cell pool. Proteins were detected by anti-GFP or anti-LBR along with an actin loading control. Expected mass of tagged gene product (LBR-FKBP-GFP) and the untagged endogenous LBR are indicated by green and black arrows, respectively. **(E)** Stills of single slices from a z-stack of live HCT116 LBR-FKBP-GFP (green) knock-in cells with SiR-DNA staining (magenta) progressing through mitosis. Time, min; scale bar, 10 µm; insets, 2× zoom. The individual values for panel B are available at https://doi.org/10.5281/zenodo.15582238.
(TIFF)

**S3 Fig. LBR-FKBP-GFP relocalization does not cluster the plasma membrane anchor or other ER proteins.** Single slices from z-stacks of live HCT116 cells co-expressing LBR-FKBP-GFP (green), LBR-mCherry **(A)** or mCherry-Sec61β **(B)** (magenta) and SH4-FRB-EBFP2, treated with rapamycin (200 nM). Scale bars, 10 µm; Insets, 4× expansion of ROI.
(TIFF)

**S4 Fig. LBR-FKBP-GFP relocalization does not alter ER morphology. (A)** Single slices of fixed HCT116 cells co-expressing LBR-FKBP-GFP (green) and SH4-FRB-EBFP2, treated with rapamycin (200 nM) for 30 min before fixation, and then stained with concanavalin A dye (PhenoVue Fluor 568-ConA, magenta). **(B)** Single slice of an HCT116 cell co-expressing LBR-FKBP-GFP (green), mCherry-Sec61β (magenta) and SH4-FRB-EBFP2, before (pre) or 7 min after addition of rapamycin (200 nM). Scale bars, 10 µm; Insets, 4× expansion of ROI.
(TIFF)

**S5 Fig. Detecting a thapsigargin-induced increase in ER-PM contact sites with relocalization of LBR-FKBP-GFP. (A)** Typical confocal images of HCT116 cells co-expressing LBR-FKBP-GFP (green), Stargazin-EBFP2-FRB (magenta), with SiR-DNA (blue in merge) to detect DNA. Cells were treated with thapsigargin (1 µM, 20 min) or not, as indicated, before relocalization was induced with rapamycin (200 nM). Scale bar, 10 µm; Insets, 4× expansion of ROI. **(B)** Plot to show the MCS fraction of the plasma membrane profile. Spot, cells; bars, mean ± sd. P-value, Student's t test. The individual values for panel B are available at https://doi.org/10.5281/zenodo.15582238.
(TIFF)

**S6 Fig. LBR-FKBP-GFP relocalizes in a similar pattern and can act as a generic marker of several ER-membrane contact types across multiple different cell types. (A)** Single slices from z-stacks of fixed RPE-1, Cos-7 and HeLa cells co-expressing LBR-FKBP-GFP (green) and Stargazin-mCherry-FRB, Mito-mCherry-FRB or FRB-mCherry-Giantin(3131-3259) (magenta), treated with rapamycin (200 nM for 30 min before fixation) where indicated and stained

with DAPI (blue). Scale bars, 10 µm; Insets, 4× expansion of ROI. **(B)** Line profiles to show the intensity of LBR-FKBP-GFP (green) and anchor protein (magenta) signal measured in a line along the structure. The individual values for panel B are available at https://doi.org/10.5281/zenodo.15582238.
(TIFF)

**S7 Fig. Relocalization of LBR-FKBP-GFP to ER-PM or ER–mitochondria contact sites using rapalog.** Single slices from z-stacks of fixed HCT116 cells co-expressing LBR-FKBP-GFP (green) and Stargazin-mCherry-FRB(T2098L) or MitoTrap (Mito-mCherry-FRB[T2098L]) (magenta) and stained with DAPI (blue). Where indicated, samples were treated with rapalog (5 µM) for 30 min prior to fixation. Scale bars, 10 µm; Insets, 6× expansion of smaller ROI or 3× expansion of larger ROI.
(TIFF)

**S8 Fig. LBR-FKBP-GFP relocalization labeling long-term. (A)** Live HeLa cells transiently expressing LBR-FKBP-GFP (green) and Stargazin-mCherry-FRB(T2098L) (magenta) with SiR-DNA staining (blue). Rapalog was added to a final concentration 5 µM after capture of images at the first time point (0 h). Images were captured at 0, 2, and 4 h. Time is indicated in hh:mm. Scale bars, 10 µm; Insets, 4× expansion of ROI. **(B)** Raincloud plot to show size distribution of LBR-FKBP-GFP clusters analyzed in 3D. **(C)** Plot of the median contact area for each cell. **(D)** Plot of the density of clusters (total clusters divided by cell surface). In C and D, dots show cells connected by a line, mean and ± sd are indicated by crossbar. The individual values for panels B, C, and D are available at https://doi.org/10.5281/zenodo.15582238.
(TIFF)

**S9 Fig. Line profiles of LBR-FKBP-GFP or FKBP-GFP-Sec61β relocalization to mark ER-lipid droplet, ER-endosome, or ER-lysosome contact sites.** Line profiles corresponding to the cells shown in Fig 5A. Plots show the intensity of LBR-FKBP-GFP or FKBP-GFP-Sec61bβ (green) and anchor protein (magenta) signal measured around the perimeter of the structure. Anchor proteins are as follows: FRB-mCherry-PLIN3 (lipid droplets), FRB-mCherry-EEA1 (early endosomes), or LAMP1-mCherry-FRB (lysosomes). Insets show the line profile images. The individual values are available at https://doi.org/10.5281/zenodo.15582238.
(TIFF)

**S10 Fig. Other proteins tested do not form clusters when relocalized to the plasma membrane or mitochondria.** Example micrographs of HCT116 cells co-expressing emerin, LAP2β or BAF construct tagged with FKBP-GFP at the N-terminus (green), with Stargazin-mCherry-FRB, Mito-mCherry-FRB (C) or a control with no FRB construct expressed (as indicated), stained with DAPI (blue). All samples were treated with rapamycin (200 nM) for 30 min prior to fixation. Shown are single slices from z-stacks of an interphase and metaphase cell. Scale bars, 10 µm; insets, 5× expansion of smaller ROI in mitochondria examples or 3× expansion of larger ROI.
(TIFF)

**S11 Fig. Line profiles of FKBP-GFP-tagged LBR constructs.** Line profiles corresponding to the cells shown in Fig 6. Plots show the intensity of FKBP-GFP-tagged LBR constructs (green) and Stargazin-mCherry-FRB (magenta) signal measured at the plasma membrane of mitotic cells. The individual values are available at https://doi.org/10.5281/zenodo.15582238.
(TIFF)

**S12 Fig. FKBP-GFP-Sec61β relocalizes to areas larger than ER-PM contact sites, even at the lowest expression levels. (A)** Single confocal micrographs of HCT116 cells expressing FKBP-GFP-Sec61β or LBR-FKBP-GFP under a CMV, PGK, crCMV or endogenous (knock-in) promoter. All images are displayed with identical contrast stretch to allow direct comparison of expression levels. **(B)** Representative confocal images of relocalization of FKBP-GFP-Sec61β to

Stargazin-mCherry-FRB (magenta) transiently expressed in HCT116 cells under CMV, PGK, or crCMV promoters. **(C)** Representative confocal images of relocalization of FKBP-GFP-Sec61β to Stargazin-EBFP2-FRB transiently expressed in HCT116 cells under CMV, PGK or crCMV promoters. Cells additionally express mScarlet-I3-6DG5-MAPPER (magenta). **(D)** Quantification of ER fluorescence of the indicated constructs in HCT116, note log scale. **(E, F)** Quantification of experiments in B to measure clusters of FKBP-GFP-Sec61β fluorescence following relocalization to Stargazin-mCherry-FRB, and to compare with equivalent analysis of LBR-FKBP-GFP, either overexpressed or in knock-in HCT116 cells. (E) Plot of the median contact area for each cell. (F) Plot of the density of clusters (total clusters divided by cell surface). Dots show cells, mean and ± sd are indicated by crossbar. *P*-values, ANOVA with Tukey's HSD test (compared to CMV). Scale bars, 10 μm; insets, 6× expansion of smaller ROI or 3× expansion of larger ROI. The individual values for panels D, E, and F are available at https://doi.org/10.5281/zenodo.15582238.
(TIFF)

**S13 Fig. Further examples of mitotic ER-Golgi contacts by SBF-SEM.** Single slices of metaphase **(A)** and telophase **(B)** HCT116 LBR-FKBP-GFP CRISPR knock-in cell SBF-SEM datasets are shown. Depth of each slice within the dataset (nm) is indicated. Example Golgi clusters are shown by yellow arrows on the full slice image. Three sequential slices of these regions (3×expansion) are shown beside. Scale bars, 2 and 0.5 μm on zoom region.
(TIFF)

**S1 Video. Example of induced relocalization of LBR-FKBP-GFP to the plasma membrane.** Movie of a mitotic HCT116 LBR-FKBP-GFP knock-in cell co-expressing Stargazin-mCherry-FRB. Rapamycin (200 nM) is added between the first and second frame. Time, mm:ss. Playback, 10 fps. Scale bar, 10 μm.
(MP4)

**S2 Video. Example of a control cell with no LBR-FKBP-GFP relocalization to the plasma membrane.** Movie of an HCT116 cell expressing LBR-FKBP-GFP (left, green) and mCherry-Sec61β (middle, red) with no anchor. Rapamycin (200 nM) is added between the first and second frame. Time, mm:ss. Playback, 5 fps. Scale bar, 10 μm.
(MP4)

**S3 Video. Example of induced relocalization of LBR-FKBP-GFP to the plasma membrane.** Movie of an HCT116 cell expressing LBR-FKBP-GFP (left, green) and mCherry-Sec61β (middle, red) with SH4-FRB-EBFP2 (blue in merge). Rapamycin (200 nM) is added between the first and second frame. Time, mm:ss. Playback, 5 fps. Scale bar, 10 μm.
(MP4)

**S4 Video. Example of LBR-FKBP-GFP relocalisation to the plasma membrane in HCT116 cells expressing MAPPER.** Movie of a mitotic HCT116 cell co-expressing LBR-FKBP-GFP (green, left), mScarlet-I3-6DG5-MAPPER (magenta, second channel), Stargazin-EBFP2-FRB (third channel) and stained with SiR-DNA (blue). Rapamycin (200 nM) is added between the first and second frame. Time, mm:ss. Playback, 2 fps. Insets, 3× expansion of ROI. Scale bar, 10 μm.
(MP4)

**S5 Video. Example of LBR-FKBP-GFP relocalisation to the plasma membrane in HeLa cells expressing MAPPER.** Movie of a mitotic HeLa cell co-expressing LBR-FKBP-GFP (green, left), mScarlet-I3-6DG5-MAPPER (magenta, second channel), Stargazin-EBFP2-FRB (third channel) and stained with SiR-DNA (blue). Rapamycin (200 nM) is added between the first and second frame. Time, mm:ss. Playback, 2 fps. Insets, 3× expansion of ROI. Scale bar, 10 μm.
(MP4)

**S6 Video. Example of induced relocalization of LBR-FKBP-GFP to mitochondria.** Movie of a mitotic HCT116 LBR-FKBP-GFP (green, left) knock-in cell co-expressing MitoTrap (Mito-mCherry-FRB, magenta, middle), stained with

SiR-DNA (blue). Rapamycin (200 nM) is added during frame 2. Time, mm:ss. Playback, 10 fps. Insets, 3× expansion of ROI. Scale bar, 10 μm.
(MP4)

**S7 Video. Example of induced relocalization of FKBP-GFP-Sec61β to mitochondria.** Movie of a mitotic HCT116 cell co-expressing FKBP-GFP-Sec61β (green, left), MitoTrap (Mito-mCherry-FRB, magenta, middle), stained with SiR-DNA (blue). Rapamycin (200 nM) is added between the first and second frame. Time, mm:ss. Playback, 1 fps. Insets, 2× expansion of ROI. Scale bar, 5 μm.
(MP4)

**S8 Video. LBR-FKBP-GFP imaging long-term in live HCT116 cells.** Live HCT116 cells transiently expressing LBR-FKBP-GFP (green, left) and Stargazin-mCherry-FRB(T2098L) (middle) with SiR-DNA staining (magenta). Videos were captured in a single z-slice at 30 min intervals. Time is indicated in hh:mm. Playback, 2 fps. Scale bar, 10 μm.
(MP4)

**S9 Video. LBR-FKBP-GFP relocalisation long-term in live HCT116 cells.** Live HCT116 cells transiently expressing LBR-FKBP-GFP (green, left) and Stargazin-mCherry-FRB(T2098L) (middle) with SiR-DNA staining (magenta). Rapalog was added to a final concentration of 5 μM between the first and second frame. Videos were captured in a single z-slice at 30 min intervals. Time is indicated in hh:mm. Playback, 2 fps. Scale bar, 10 μm.
(MP4)

**S10 Video. Golgi HCT116 video.** Movie of a mitotic HCT116 cell co-expressing LBR-FKBP-GFP (green, left), FRB-mCherry-Giantin3131-3259 (magenta, middle) and stained with SiR-DNA (blue). Rapamycin (200 nM) is added between the first and second frame. Time, mm:ss. Playback, 2 fps. Insets, 3× expansion of ROI. Scale bar, 10 μm.
(MP4)

**S1 Raw Images. Uncropped blots and gel from S2 Fig.**
(PDF)

## Acknowledgments

We thank Maëlle Lorvellec and Laura Cooper from the Computing and Advanced Microscopy Unit (CAMDU) for their help and support. Alex Moore provided technical assistance to make the LBR-FKBP-GFP knock-in cells and to manually segment SBF-SEM data for nnU-Net training. Alison Beckett at the Liverpool Biomedical EM Unit carried out SBF-SEM imaging. We also acknowledge the help of Steven Servin-Gonzalez and the use of the Flow Cytometry Shared Resource Laboratory at the University of Warwick. We are grateful to members of the Royle lab for feedback on the project and manuscript.

## Author contributions

**Formal analysis:** Laura Downie, Stephen John Royle.

**Investigation:** Laura Downie, Nuria Ferrandiz, Elizabeth Courthold, Megan Jones.

**Methodology:** Laura Downie.

**Software:** Stephen John Royle.

**Visualization:** Laura Downie, Stephen John Royle.

**Writing—original draft:** Laura Downie, Stephen John Royle.

**Writing—review & editing:** Laura Downie, Elizabeth Courthold, Stephen John Royle.

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
