## [Editor Report · Decision Letter 0]

Dear Dr Royle,

Thank you for submitting your revised manuscript entitled "Non-disruptive inducible labeling of ER-membrane contact sites using the Lamin B Receptor" for consideration as a Methods and Resources Article by PLOS Biology. Please accept my apologies for the delay in getting back to you with feedback as we consulted with the original Academic Editor about your submission.

Your revision and rebuttal has now been evaluated by the PLOS Biology editorial staff and I am writing to let you know that we would like to send your submission back to the original reviewers out for external peer review.

Once your full submission is complete, your paper will undergo a series of checks in preparation for peer review. After your manuscript has passed the checks it will be sent out for review. To provide the metadata for your submission, please Login to Editorial Manager (https://www.editorialmanager.com/pbiology) within two working days, i.e. by Nov 17 2024 11:59PM.

Kind regards,

Richard

Richard Hodge, PhD

rhodge@plos.org

PLOS

---

## [Decision Letter · Decision Letter 1]

Dear Dr Royle,

Thank you for your patience while we considered your revised manuscript "Non-disruptive inducible labeling of ER-membrane contact sites using the Lamin B Receptor" for publication as a Methods and Resources article at PLOS Biology. Please accept my sincere apologies for the delays that you have experienced during this round of the peer review process. Your revised study has been evaluated by the PLOS Biology editors, the Academic Editor and all four of the original reviewers.

In light of the reviews, which you will find at the end of this email, we would like to invite you to revise the work to thoroughly address the reviewers' reports.

As you will see below, the reviewers appreciate the additional work included in the revised version and generally agree that many of the previous comments have been addressed. However, the reviewers continue to raise some overlapping concerns about the strength of the methodological validation and demonstration of utility. This includes the additional data included to show that the method does not simply work due to finding a ‘Goldilocks’ zone of LBR expression and Reviewer #3 notes that a careful quantification of ER levels of LBR vs Sec61 is needed to support the claims. Further to this, Reviewer #1 notes that the new data is not quantified to the same standard as the earlier data and Reviewer #2 notes that LBR puncta should be quantified relative to the targeted organelle at different time-points after rapamycin. In addition, Reviewer #2 would like to see a comparison of LBR relocalisation and Sec61β and in the same cell.

Given the extent of revision needed, we cannot make a decision about publication until we have seen the revised manuscript and your response to the reviewers' comments. Your revised manuscript is likely to be sent for further evaluation by all or a subset of the reviewers.

**IMPORTANT - SUBMITTING YOUR REVISION**

*Re-submission Checklist*

*Published Peer Review*

*PLOS Data Policy*

*Blot and Gel Data Policy*

Best regards,

Richard

Richard Hodge, PhD

rhodge@plos.org

REVIEWS:

Reviewer #1: In this revised manuscript, Downie et al. have done substantial work to address the reviewer's criticisms. The main criticism however remains: they propose a new method to label ER contact sites, but have no clue of how it works. Briefly, they use a rapamycin-dimerizable pair of proteins, one targeted to the ER, and one targeted to another organelle. They observe that using either full-length of fragments of LBR as ER-targeted moiety outperforms using sec61beta, and presumably (although not tested with the same rigor) other ER proteins. Yet, they cannot rationalize why.

Major points:

1-Not being able to rationalize the data appears like a major shortcoming of the study. One could argue that, as long as it works, we do not have to care how it works. I disagree; to be useful, this method will need to be adaptable to the precise question that any user will want to put to it. Not knowing how it works makes it impossible how to adapt the method to ensure best results in a variety of scenarios. Perhaps, the authors could re-write their paper, not as a method, but as an observation: different ER-targeting moieties have different behaviours when they are artificially targeted to contact sites. Yet, with no physiological significance and no mechanistic insight, this observation appears orphaned. One way to obtain a beginning of a mechanistic insight could be to study in detail the (1-245) N terminal fragment that the authors claim is sufficient for the phenomenon. They could for instance exchange the N-terminal linker and the TM domains to figure which one is actually important for the phenomenon, and test possible models by mutagenesis.

2-I have raised the possibility that expression levels were responsible for the different behaviours of the tested constructs. The authors here provide additional data, where they express more or less of LBR-based constructs and sec61b-based constructs. Their conclusions is that it is the nature and not level of the protein that does the trick. Again, I beg to differ. In fig S12, the crippled CMV promoter driving low level of sec61-based construct solves pretty much all the problems of the sec61-based constructs. It does not increase the staining extent of the MAPPER reporter, indicating that it does not visibly increase the extent of ER-PM contacts. Also the protein itself colocalizes perfectly with MAPPER (the boxes that the authors highlight are, in that respect, not truly representative of the rest of the image), as seen with the LBR-based constructs. Therefore, while it likely does not explain 100% of the difference in the staining between LBR- and sec61-based reporter, expression level appears to be a major determinant of what does the trick.

3-The authors have performed substantial work to answer the reviewers criticisms. One issue however is that these data are not quatified to the same standard. The data in their first version is precisely quantified and compared with each other. This care is commendable, but the need for such careful quantification also reflects the subtle quantitative differences (or absence thereof) reported here. The added new data is however not performed at the same standard. For instance,

-the data on the differentially expressed constructs is not quantified and relies on visual inspection of micrographs, which as we have seen before, depends on the observer (my interpretation of the data on the crippled CMV promoter is opposite to the author's).

-The additional cell lines that are used to show that the approach can be generalized are, again, not quantified.

-Whether the single image of GFP-FKBP-LBR (Fig 6C) upon rapamycin looks more like LBR-FKBP-GFP (Fig 6B, S1) or like FKBP-GFP-Sec61 (Fig 1B, S1), is not obvious and left to the appreciation of the reader. My appreciation would again contradict the author's conclusions.

4-The Thapsigargin data are disappointing if the authors want to argue that their probe is sensitive to changes in ER-PM contacts. TG treatment is expected to multiply the extent of ER-PM contacts by 2 to 10 (https://www.pnas.org/doi/10.1073/pnas.0911280106). The cited study utilizing MAPPER sees a 2-fold increase in MAPPER intensity. The ~20% increase reported here indicates that LabelER is not a great reporter for ER-PM contacts dynamics, or that the stimulus did not yield the expected increase in membrane apposition. It might be necessary to TG-treat cell in calcium-free medium if one wants to stimulate SOCE the 20-minutes necessary here. Anyways, to make a decisive point here, it is necessary to quantify the increased ER-PM apposition with another method (e.g. using the MAPPER, for which a good benchmark exists).

Reviewer #2: The manuscript is much improved and many of my concerns have been addressed. However, the field has progressed during the revision period and in light of the recent report by Zhao et al (Cell Reports 2024) demonstrating a role for LBR as a tether connecting IP3R on the ER with VDAC2 on mitochondria to mediate mitochondrial calcium influx, the targeting of LBR to specific organelles seems likely to affect calcium homeostasis. This, coupled with intrinsic irreversibility, raises a question mark over the value of induced LBR relocalisation to the community as a tool to study MCS, which are often both important for and regulated by cellular calcium. Nevertheless a lot of work has gone into the study and some very nice data is presented. The revisions have improved the quality of the manuscript but I still have a few concerns:

* Global effects of induction of LBR relocalisation on the ER: The authors claim that 3D-EM reconstruction of ER networks by SBF-SEM in multiple cells with LaBeRling of mitochondria- ER MCSs vs controls (Figure 4) didn't reveal any gross changes in ER morphology. The 3D-EM is a great addition to the paper, but I'm not seeing the analysis of ER morphology in Figure 4 - maybe I'm overlooking something? In the examples shown in A, there seems to be quite a striking difference in ER distribution in control v rapamycin-treated cells (SBF-SEM panels).

Similarly, I understand that LBR relocalisation has been compared with that of Sec61β but would have liked to see that in the same cell - ie, does relocalisation of LBR to MCS following rapamycin affect ER morphology as visualised by other ER markers (in the same cell)? Fig S3B goes part way to address this, but without the no rapamycin control it's very hard to determine if ER distribution is affected. To me, the data shown appears to suggest that there is some rearrangement of the ER network with a high amount of Sec61 staining at the PM, but it's hard to tell if this is increased without seeing Sec61 distribution in the corresponding no rapamycin control cells. The new data in FigS4 is helpful but not many conclusions can be drawn from an image of a single cell, also with no controls.

* Irreversibility: New data showing the effect of stabilising MCS (FigS8) is encouraging but it should be stated clearly that the LaBeRling system is irreversible rather than alluding to it vaguely in the final sentence (that contains a typo for membrane).

5. Figure 3B LBR puncta increase with time. The authors' assertion that green signal weakens over time as a result of photobleaching does not match what I see on the figure. Although the signal in the magnified area is reduced at 9 min, in the cell as a whole, the LBR puncta appear to be increased both in number and intensity. In contrast, the mitotrap signal has faded with photobleaching, so quantitation of LBR puncta:mito signal ratio would probably reveal a significant relative increase in LBR intensity. I would like to see quantitation of LBR puncta relative to the targeted organelle in large fields of view imaged at different (including longer) time-points after rapamycin. This doesn't need to be continuous imaging, or the same individual cell - snapshots in large numbers of cells at different timepoints could be used for quantitation. 3D adds a nice measurement of MCS induced by LBR v Sec61 but as far as I can tell only 3 cells were included in the analysis for control cells (4 for +Rapamycin) which, given the cell-to-cell variability in Figure 4D, is a very low number for meaningful quantitation. Also the scatter plots suggest increased "long" tethering (>30nm) of ER-mitochondria contacts, whereas a 20nm inter-organelle gap was shown to be optimal for IP3R-dependent Ca2+ transfer at the MCS, with reduced Ca2+ transfer at 30nm distnces (Dematteis et al, Commun Biol, 2024), suggesting that the LRN-FKBP-GFP construct may not be functional at the IP3R-VDAC2 complex. In terms of its potential utility as a MCS sensor, it would be important to know if it affects Ca2+ flux.

Reviewer #3: The authors have made some impressive improvements and meaningful experimental additions to the manuscript. I am convinced by most of their amendments with a few important exceptions below. Should they address these remaining concerns I would support publication of the work.

"We were aware of the split-FAST technology, but we had missed its application to contact sites ("PRINCESS" and "FABCON" methods). We now reference both of these papers in the Introduction. There are advantages and disadvantages to all methods. We note the benefit of reversibility of split-FAST. We would counter that while this is an advantage, a downside of split-FAST is that it is proprietary tech. Our method doesn't rely on purchase of materials from "The Twinkle Factory". Researchers can grab our constructs off Addgene, get hold of rapamycin from any supplier, and they are good to go. Since we haven't used split-FAST we are not in a position to compare it with our method directly."

The authors have now included the appropriate references and their justification above is reasonable. However, this justification should be made clear in the manuscript e.g. these approaches rely on proprietary costly fluorophores. In its current writing, it seems as if the authors suggest that split-FAST also artificially increases contacts, which is not the case based on the data from the PRINCESS and FABCON papers.

"As the Reviewer notes, our intention is for the method to be used short-term, but we were happy to investigate this point in case users wondered if LaBeRling can be used on a longer time scale. Due to the issues with long-term application of rapamycin, we show that rapalog AP21967 can be used as an alternative with the T2098L mutation in FRB (Supplementary Figure S7). Using this regime we show:

1. Long-term LaBeRling of PM-ER MCSs with rapalog (Supplementary Figure S8)

2. No effect on cell migration or viability under conditions of long-term LaBeRling (up to 8.5 h) of PM-ER MCSs with rapalog (Supplementary Videos 6 and 7)"

These are useful additions to the manuscript and although only demonstrated for one contact site, this seems like an adequate response.

"This point was also the main criticism raised by Reviewer 1. Firstly, we have now analyzed the expression of LBR-FKBP-GFP and FKBP-GFP-Sec61β by western blotting (Supplementary Figure S11). It is true that the expression of LBR-FKBP-GFP is lower than that of FKBP-GFP-Sec61β, however the expression of the truncated LBR constructs is higher than that of Sec61β, and since these can label MCSs, it argues against the idea that labeling is solely due to lower ER expression. We know that overexpression of LBRFKBP-GFP doesn't disrupt the contacts (already shown in the paper). However, the counter-view is that even under conditions of higher expression, because the nuclear envelope pool of LBR may be anchored, the ER pool could still be less that that of Sec61β. Secondly, to tackle this, we reduced the expression of FKBP-GFP-Sec61β using either a PGK or a crippledCMV promoter. Despite the lower levels, distortion of PM-ER contacts was still seen (Supplementary Figure S12). These data argue against a scenario where LBR has a goldilocks expression level where it is low enough to be used in this way, and not high enough to distort contacts. Even if this is the case, we can say that the range of expression that allows LaBeRling is very wide. In indirect support of this, we have also added new data to demonstrate that LaBeRling is possible in three other cell lines (RPE-1, Cos-7 and HeLa), which further argues against the possibility that the application of LBR is limited by expression levels in one cell line. The purpose of our paper is two-fold: (i) to show that LBR can be used to label ER MCSs and (ii) to report the existence of Golgi-ER contacts during mitosis. A secondary aspect to the paper is why LBR can be used in this way. The answer to this question doesn't really matter for the two central messages of our paper."

I'm afraid I don't find this additional data very convincing. As the authors recognise, the critical issue here is the protein levels of ER localised LBR, not total LBR (thus the WB does not really help sine it is ER+NE LBR levels). Likewise, the authors do not provide evidence that the crippled CMV or PGK promoter reduce ER levels of Sec61 to comparable ER levels of LBR. I propose two solutions, one is to carefully quantitate the ER levels of LBR vs Sec61 using the microscopy data collected (given the FP is identical and effects of self-quenching/folding should be comparable). The alternative simpler solution is to state clearly in the manuscript that this may indeed be a goldilocks expression level phenomena but it still works! I would find either approach acceptable, although of course the former is more intellectually satisfying.

"We were not precise enough here. We meant "governed by proteins on the ER side". This was implied by the preceding sentence, but the Reviewer is correct to pull us up that we seemed to be saying the ER itself was important. The Obara et al. paper nicely demonstrates this concept because the ALS mutation in VAPB which changes diffusivity, alters the mitochondria-ER MCS and supports our point. We now cite this paper in the Discussion."

Apologies if my point wasn't made clearly the first time. I understood the authors initial prose, but instead was surprised by the assertion that the ER tethering partner governs contact sites and that the other organellar tethering partner may be less important. Figure 3 from the Obara et al., manuscript suggests that overexpression of VAPB (the ER partner) has very little effect on ERMCS abundance, while overexpression of PTPIP51 dramatically increases contact sites. This has been shown with other similar systems, where the ER tether abundance seems to have minimal effect, while the abundance of FFAT containing organellar partners seems to have a dramatic effect. As such, I would caution the authors against interpreting that "ER-MCSs are not truly bipartite and are perhaps governed by the ER".

"This is a misunderstanding: MAPPER is not rapamycin-inducible. In the original paper there is an FRB domain in the MAPPER construct (although this is not used for inducing labeling of contact sites) but the construct is constitutively at PM-ER MCSs. In our paper, we removed the FRB domain and replaced it with a similar structured protein (with PDB code 6DG5). We call our version of this construct mScarlet-I3-6DG5-MAPPER, and have made it available on Addgene since the original MAPPER construct is not in the public domain. We had to make this construct because we struggled to find antibodies or other constructs that worked for specifically labeling MCSs (perhaps we were just unlucky).Nonetheless, using something like MAPPER is a concern that we would alter PM-ER MCSs. We included the control of no MAPPER co-expression to assess the impact of expression. The suggestion to modulate contacts is a good one. We used thapsigargin treatment to expand PMER contact sites, as described in PMID: 24183667 (the original MAPPER paper). Following thapsigargin treatment we could indeed see more extensive PM-ER MCSs with LaBeRling (Supplementary Figure S5). We think this is a nice addition to the paper which shows that we can detect changes in PM-ER MCSs with LaBeRling."

Thank you for clarifying this point and apologies for the misunderstanding. I agree that the thapsigargin experiment is a strong addition to the paper and my initial point has been well addressed.

All my remaining concerns have been sufficiently addressed.

Reviewer #4: I thank the authors for their very thoughtful responses to the concerns raised. The authors have adequately addressed the concerns I had with the initial manuscript. They have strengthened the data to support that the LaBeRling method can be used to non-disruptively label ER MCSs in cells, which will prove useful to the MCS field.

One minor comment is regarding the changes made to Figure 3A in response to comments that both I and Reviewer 2 made. Using different scaling on the images is acceptable. However, the authors should state in the legend or methods that different scaling was used between panels and direct comparisons of intensity cannot be made.

---

## [Decision Letter · Decision Letter 2]

Dear Stephen,

Thank you for your patience while we considered your revised manuscript "Non-disruptive inducible labeling of ER-membrane contact sites using the Lamin B Receptor" for publication as a Methods and Resources Article at PLOS Biology. Please accept my apologies for the delays that you have experienced during this round of the peer review process. This revised version of your manuscript has been evaluated by the PLOS Biology editors, the Academic Editor and the original reviewers.

Based on the reviews, I am pleased to say that we are likely to accept this manuscript for publication, provided you satisfactorily address the remaining points raised by the reviewers. Please also make sure to address the following data and other policy-related requests that I have provided below (A-F):

(A) You may be aware of the PLOS Data Policy, which requires that all data be made available without restriction: http://journals.plos.org/plosbiology/s/data-availability. For more information, please also see this editorial: http://dx.doi.org/10.1371/journal.pbio.1001797

-Supplementary files (e.g., excel). Please ensure that all data files are uploaded as 'Supporting Information' and are invariably referred to (in the manuscript, figure legends, and the Description field when uploading your files) using the following format verbatim: S1 Data, S2 Data, etc. Multiple panels of a single or even several figures can be included as multiple sheets in one excel file that is saved using exactly the following convention: S1_Data.xlsx (using an underscore).

-Deposition in a publicly available repository. Please also provide the accession code or a reviewer link so that we may view your data before publication.

Figure 1D-E, 1G, 1I, 2B-C, 3C-D, 4C-D, 5B, 7B, S2B, S5B, S6B, S8B-D, S9, S11, S12D-F

(B) Please also ensure that each of the relevant figure legends in your manuscript include information on *WHERE THE UNDERLYING DATA CAN BE FOUND*, and ensure your supplemental data file/s has a legend.

(C) We require the original, uncropped and minimally adjusted images supporting all blot and gel results reported in the following Figures:

Figure S2C-D

We will require these files before a manuscript can be accepted so please prepare and upload them now. Please carefully read our guidelines for how to prepare and upload this data: https://journals.plos.org/plosbiology/s/figures#loc-blot-and-gel-reporting-requirements

(D) When searching for the code deposited in Github, the link provided in the Data Availbiltiy Statement did not work for us (https://github.com/quantixed/p65p038). Could you please check that the URL provided is correct?

In addition, please note that we cannot accept sole deposition of your code in GitHub, as this could be changed after publication. However, you can archive this version of your publicly available GitHub code to Zenodo. Once you do this, it will generate a DOI number, which you will need to provide in the Data Accessibility Statement (you are welcome to also provide the GitHub access information). See the process for doing this here: https://docs.github.com/en/repositories/archiving-a-github-repository/referencing-and-citing-content

(E) Please note that per journal policy, we do not allow the mention of "data not shown", "personal communication", "manuscript in preparation" or other references to data that is not publicly available or contained within this manuscript. Please either remove mention of these data in the figure legends (Figure 2, 4, S3, S4, SV1) or provide figures presenting the results and the data underlying the figure(s).

(F) Please note that per journal policy, the model system/species studied should be clearly stated in the abstract of your manuscript.

We expect to receive your revised manuscript within two weeks.

*Published Peer Review History*

*Press*

Best regards,

Richard

Richard Hodge, PhD

rhodge@plos.org

Reviewer remarks:

Reviewer #1: The amount of new work by the authors is highly commendable, and I don't think further work can help make their case.

I unfortunatley remain unconvinced of the importance and usefulness of this tool.

1-Reducing the expression level appears to do most of the trick, even if not correlating with absolute amounts of protein (see point 2). Somehow, using the BFP- rather than mCherry-tagged Stargazin also helps(figure s12C). The sec61b foci of figure S12C (PGK and crCMV) appear more discrete than those of GFP-FKBP-LBR in figure 6C, for instance. The sentence "reducing the expression of other ER proteins didn't allow for specific labeling" is misleading; again, I see no difference in the specificity of labelling in fig S12C PGK and crCMV and other data using LBR or fragments thereof (the quantified data in S12D-F is for S12B, not S12C).

2-Explaining why LBR works differently to sec61 appears paramount to generate a useful method. The authors do not agree, so I will illustrate my point. A reporter such as the one described here will induce artifacts because of the high affinity of the FKBP-FRB interaction, and the elasticity of membranes; if more proteins are available to engage in interaction, then new interactions will be formed, which will increase the tethering between the organelles, and thus the surface of contact. The only way to limit artifacts is to reduce the number of proteins that can engage in an interaction. There are two ways of reducing the number of proteins involved in the interaction. First is to reduce the absolute protein level (see point 1), and 2nd is to reduce the availability of the protein for interaction, including (but not limited to), for instance, reducing its ability to diffuse towards contact sites. Now what could specifically reduce LBR's availability for interaction? According to the authors, the 1st 254 aminoacids only are necessary to do the trick. In these 254 aminoacids we find the transmembrane domain (of course) and a perfectly folded and functional Tudor domain (e.g. PMID:22052904). Tudor domains bind acetylated lysines through pi-stacking, and the aromatics necessary are conserved in LBR. This is likely, together with lamin binding, one reason for the concentration of LBR in the inner nuclear membrane (binding methylated histones in the chromatin). In the cytosol, LBR's Tudor domain might bind other methylated residues, on ribosomes, spliceosomes, Hsp90 and such, or phase separate into nanodomains, as other Tudor proteins have been shown to do (PMID:34115980). This might be what limits the amount of LBR molecules able to engage in contact-site interactions. That might be a very good thing until it becomes a problem.

As this is a method paper, it directs future users to leverage a tried a true method for assessing contact sites amounts and localization in a way that is "non-disruptive and does not change ER-MCSs", in this or that experimental conditions. Now "this or that" situation (be it a drug treatment, a genetic alteration, a metabolic or differentiation status) might very well affect the degree of lysine/arginine methylation, the phase separation of tudor domains, chromatin structure, Ribosomal proteins PTMs, etc. which will, in turn change the availability of LBR, and hence the amount of contact site detected by the method. This is why it is much more desirable to know what controls the availability of proteins available to engage in contact sites (e.g. by modulating expression level), rather than relying on an unknown process. Now of course this whole story of tudor domains is invented and might be wrong. The authors might find that a tudor-domain mutant works just as well. But we still wont not know what makes the reporter work, and whether "this or that" situation affect contact sites, or affect the availability of LBR through other mechanisms.

3-The emerin mitochondrial foci (S10) appear no different from the LBR ones (3A). The Lap2b ER-PM foci (S10) appear no different from the LBR ones (6B top, 6C top). This might be related to points 1 and 2 in that these proteins might be in the sweet-spot for expression and availability.

Reviewer #2: Concerns have mostly been addressed but I think the authors should elaborate a little bit in the discussion about the potential role of LBR in generating platforms for Ca2+ flux. If researchers want to use this approach as a contact sites sensor, it's important they are aware of likely impact on calcium signaling.

Reviewer #3: The authors have done an excellent job of addressing my remaining concerns, and I think that the addition of figureS12, in particular, is an important inclusion to the work. I am happy to support publication of the manuscript and congratulate the authors on their efforts.

Reviewer #4: The reviewers have adequately addressed the concerns I raised.

---

## [Editor Report · Decision Letter 3]

Dear Stephen,

On behalf of my colleagues and the Academic Editor, Frederick Hughson, I am pleased to say that we can accept your manuscript for publication, provided you address any remaining formatting and reporting issues. These will be detailed in an email you should receive within 2-3 business days from our colleagues in the journal operations team; no action is required from you until then. Please note that we will not be able to formally accept your manuscript and schedule it for publication until you have completed any requested changes.

PRESS

Best wishes, 

Richard

Richard Hodge, PhD

rhodge@plos.org

PLOS
